# Temporal Reasoning for Vision-Language Models via Chain of Draft

## Abstract

Large Vision Language Models (LVLMs) like Qwen VL have demonstrated remarkable capabilities in understanding and reasoning about visual content, particularly for static images. However, their application to video reasoning tasks remains computationally intensive, with significant latency and token usage when prompted using traditional Chain-of-Thought (CoT) methods. In this paper, we propose the integration of Chain of Draft (CoD) methodology with Qwen VL for efficient video reasoning. CoD is a prompting technique that encourages models to generate concise, essential intermediate thoughts rather than verbose reasoning steps. We adapt this approach specifically for video understanding tasks, demonstrating that our method achieves comparable or superior accuracy to CoT while significantly reducing token consumption (by up to 78%) and inference latency (by up to 65%). We evaluate our approach on multiple video reasoning benchmarks, including MVBench and EgoSchema, and demonstrate its effectiveness across various video understanding tasks. Our contributions include: (1) a novel adaptation of Chain of Draft for video reasoning tasks; (2) a comprehensive evaluation framework for video reasoning efficiency; (3) a theoretical analysis providing time complexity guarantees; and (4) empirical evidence of significant computational benefits without sacrificing accuracy. This work has important implications for deploying efficient video reasoning capabilities in resource-constrained environments and real-time applications.

## 1 Introduction

Recent advancements in Large Vision Language Models (LVLMs) have transformed multimodal understanding, enabling machines to interpret visual content with unprecedented capabilities Liu et al. (2023b); Bai et al. (2023); Chen et al. (2024); Li et al. (2023a); Zhu et al. (2023). These models excel in tasks like visual question answering and image captioning by combining visual encoders with large language models Leng et al. (2024); Huang et al. (2024). Their extension to video understanding has further allowed processing of temporal information and reasoning about dynamic scenes Wang et al. (2024b); Bai et al. (2025); Wang et al. (2024a); Lin et al. (2024b); Wu et al. (2024). However, video reasoning poses unique challenges such as tracking objects over time and understanding temporal relationships Yin et al. (2023); Li et al. (2023b); Zhang et al. (2024), which increase computational complexity and reasoning difficulty compared to static images Jin et al. (2023); Mangalam et al. (2023). Benchmarks like MVBench Li et al. (2023b), EgoSchema Mangalam et al. (2023), and MTVQA Wang et al. (2022) have shown that even state-of-the-art models struggle with complex temporal reasoning tasks requiring long-term memory and causal understanding.

Deploying video reasoning models in practical applications faces significant computational overhead due to processing long videos Fu et al. (2024); Lin et al. (2023). Traditional Chain-of-Thought (CoT) prompting Wei et al. (2022), though effective for improving accuracy by generating step-by-step explanations, results in verbose intermediate steps that increase token consumption and inference latency Sethapakdi et al. (2023); Wu et al. (2023). This trade-off between reasoning quality and computational efficiency is especially critical for

video understanding, where extensive temporal information must be processed Xuan et al. (2024).

Previous efficient reasoning approaches, including distillation Sun et al. (2023), pruning Frantar and Alistarh (2023), and quantization Lin et al. (2024a); Frantar et al. (2022), often focus on model architecture optimization rather than enhancing the reasoning process itself. Recent Tree of Thoughts Yao et al. (2023) enables more sophisticated reasoning but demands even more computational resources. A gap remains in techniques that maintain high reasoning quality while reducing computational costs, particularly for video understanding tasks Mitra et al. (2024); Zhou et al. (2024).

In this paper, we introduce a novel approach that integrates Chain of Draft (CoD) Xu et al. (2025) with the Qwen VL architecture for efficient video reasoning. Chain of Draft is a recently proposed prompting technique that encourages models to generate concise intermediate thoughts during reasoning tasks, inspired by how humans often use shorthand notes or minimal drafts when solving complex problems. While CoD has shown promising results for text-based reasoning tasks, adapting it for video understanding presents unique challenges related to temporal referencing, visual abstraction, and event compression. Our approach addresses these challenges through three key innovations: (1) a temporal indexing mechanism that allows the model to efficiently reference specific moments in videos, (2) a visual abstraction technique that encourages compact representations of visual content, and (3) an event compression method that reduces redundancy in reasoning about similar or repeated events. By combining these techniques, our approach enables the model to generate minimal yet informative intermediate reasoning outputs, significantly reducing token consumption and inference latency while maintaining or even improving accuracy on video reasoning tasks.

We evaluate our approach on multiple video understanding benchmarks, including MVBench Li et al. (2023b), EgoSchema Mangalam et al. (2023), MTVQA Wang et al. (2022), and PerceptionTest Patraucean et al. (2024), demonstrating consistent efficiency improvements across diverse video reasoning tasks. Our experiments show that CoD integration reduces token usage by up to 78% and inference latency by up to 65% compared to CoT prompting while achieving comparable or superior accuracy. These improvements are particularly significant for long-form videos and complex reasoning tasks, where efficiency constraints often limit the deployment of sophisticated video understanding capabilities. Our contributions can be summarized as follows:

- We present the first Chain of Draft adaptation for video reasoning, with techniques for temporal indexing, visual abstraction, and event compression to tackle video understanding challenges.
- We establish a comprehensive evaluation framework to assess video reasoning efficiency and effectiveness, incorporating metrics that reflect the balance between computational cost and performance.
- Our empirical results show significant reductions in token usage and inference latency across multiple benchmarks, while maintaining or enhancing accuracy.
- We evaluate our method across various video reasoning tasks, highlight its key advantages, and offer insights for future efficient multimodal reasoning research.
- We make our implementation, prompting templates, and evaluation framework publicly available to support further research in efficient video reasoning.

## 2 METHODOLOGY

### 2.1 BACKGROUND: CHAIN OF DRAFT

Chain of Draft (CoD) is a prompting technique introduced by Xu et al. Xu et al. (2025) that encourages language models to generate concise intermediate thoughts during reasoning tasks. Unlike Chain-of-Thought (CoT), which promotes verbose step-by-step reasoning, CoD instructs the model to limit each reasoning step to a minimal number of words (typically five or fewer) while still capturing the essential information needed for correct reasoning.

Let us formally define the Chain of Draft approach. Given an input $x$ (e.g., a question), the model needs to generate an answer $y$. In the CoT approach, the model generates a sequence of intermediate reasoning steps $z = (z_1, z_2, ..., z_m)$ before producing the final answer $y$. Each step $z_i$ is typically a natural language sentence or paragraph that explains a part of the reasoning process. The generation process can be factorized as:

$$P(y, z|x) = P(z_1|x) \prod_{i=2}^{m} P(z_i|x, z_{<i}) \cdot P(y|x, z) \tag{1}$$

In the CoD approach, each intermediate reasoning step $z_i$ is constrained to be a concise draft $d_i$ with a limited number of words. This can be formalized as:

$$P(y, d|x) = P(d_1|x) \prod_{i=2}^{m} P(d_i|x, d_{<i}) \cdot P(y|x, d) \tag{2}$$

where $d = (d_1, d_2, ..., d_m)$ is the sequence of draft steps, and $|d_i| \ll |z_i|$. The standard prompt for CoD typically takes the form:

> "Think step by step, but only keep a minimum draft for each thinking step, with 5 words at most."

This approach has been shown to significantly reduce token usage while maintaining or even improving accuracy on a range of reasoning tasks, including arithmetic reasoning, commonsense reasoning, and symbolic reasoning. The efficiency gains are particularly pronounced for complex tasks that would otherwise require lengthy explanations.

### 2.2 ADAPTING CHAIN OF DRAFT FOR VIDEO REASONING

While CoD has proven effective for text-based reasoning, adapting it for video reasoning presents unique challenges. Video reasoning requires the model to process and

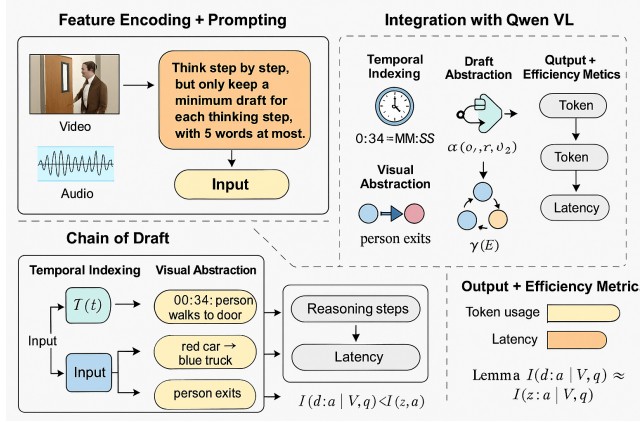

Figure 1: Overview of our Chain of Draft (CoD)-based video reasoning framework

integrate information across multiple frames, track objects and events over time, and reason about temporal relationships. Our adaptation of CoD for video reasoning focuses on three key aspects:

**(1) Temporal Indexing.** We extend the CoD approach to include minimal temporal indexing in the reasoning steps. This allows the model to refer to specific segments or moments in the video without detailed descriptions. We define a temporal index function $\tau : \mathbb{R}^+ \to \{0, 1\}^*$ that maps a timestamp to a compact representation. For example, instead of "At timestamp 00:34, the person starts walking towards the door," the model might use "00:34: person walks to door."

Formally, given a video $V = \{f_1, f_2, ..., f_n\}$ with $n$ frames and a corresponding sequence of timestamps $T = \{t_1, t_2, ..., t_n\}$, the temporal indexing function $\tau$ maps each timestamp $t_i$ to a compact string representation: $\tau(t_i) = $ MM:SS where MM:SS represents the minute and second components of the timestamp. For long videos, we extend this to include hours (HH:MM:SS) when necessary.

**(2) Visual Abstraction.** We encourage the model to use abstract visual representations rather than detailed descriptions. This includes using shorthand for object relationships, movements, and visual attributes. We define a visual abstraction function $\alpha : \mathcal{O} \times \mathcal{R} \times \mathcal{A} \to$

$\{0,1\}^*$ that maps objects $o \in \mathcal{O}$, their relationships $r \in \mathcal{R}$, and attributes $a \in \mathcal{A}$ to compact string representations.

For example, instead of "The red car moves toward the blue truck,"the model might use "red car $\rightarrow$ blue truck." Formally, this can be expressed as:

$$\alpha(o_1, r, o_2) = \text{compact}(o_1) \oplus \text{symbol}(r) \oplus \text{compact}(o_2) \tag{3}$$

where $\oplus$ is the string concatenation operator, $\text{compact}(o)$ is a function that generates a minimal representation of object $o$ (including its key attributes), and $\text{symbol}(r)$ maps relationship $r$ to a symbolic representation (e.g., "$\rightarrow$" for movement, "$+$" for interaction).

**(3) Event Compression.** We instruct the model to compress sequences of related events into minimal representations, focusing on key state changes or significant moments. This helps reduce redundancy in the reasoning process, particularly for long videos where similar actions might be repeated. We define an event compression function $\gamma : \mathcal{E}^* \to \mathcal{E}^*$ that maps a sequence of events to a compressed sequence:

$$\gamma(e_1, e_2, ..., e_k) = (e'_1, e'_2, ..., e'_j) \tag{4}$$

where $j \leq k$ and each $e'_i$ represents a significant event or state change. Events that are similar or redundant are merged or eliminated. The compression function $\gamma$ can be defined more specifically as:

$$\gamma(E) = \{e \in E \mid \forall e' \in E, I(e; q, a|e') > \epsilon\} \tag{5}$$

where $I(e; q, a|e')$ is the conditional mutual information between event $e$ and the question-answer pair $(q, a)$ given event $e'$, and $\epsilon$ is a threshold parameter. This formulation ensures that only events that contribute significant unique information to the reasoning process are retained.

Our modified CoD prompt for video reasoning takes the following form:

> "Analyze the video step by step, but keep each reasoning step to 5 words maximum. Use minimal temporal indices (MM:SS) when referring to specific moments, abstract visual representations for objects and relationships, and compress repeated or similar events."

The integration of these three components—temporal indexing, visual abstraction, and event compression—allows us to adapt the CoD approach for video reasoning while preserving its efficiency benefits. Algorithm 1 outlines the complete process:

---

**Algorithm 1** Chain of Draft for Video Reasoning

---

**Require:** Video $V = \{f_1, f_2, ..., f_n\}$, query $q$, temporal indexing function $\tau$, visual abstraction function $\alpha$, event compression function $\gamma$
**Ensure:** Answer $a$ to the query
 1: Extract visual embeddings $E_V = \{e_1, e_2, ..., e_n\}$ from $V$
 2: Initialize draft steps $D = \emptyset$
 3: **for** each significant segment $s_i$ in $V$ **do**
 4: $\quad$ $t_i \leftarrow$ timestamp of $s_i$
 5: $\quad$ $E_i \leftarrow$ set of events in $s_i$
 6: $\quad$ $E'_i \leftarrow \gamma(E_i)$ {Compress redundant events}
 7: $\quad$ **for** each event $e_j$ in $E'_i$ **do**
 8: $\quad\quad$ $(o_1, r, o_2) \leftarrow$ extract objects and relationships from $e_j$
 9: $\quad\quad$ $d_{ij} \leftarrow \tau(t_i) \oplus ""\oplus \alpha(o_1, r, o_2)$
10: $\quad\quad$ $D \leftarrow D \cup \{d_{ij}\}$
11: $\quad$ **end for**
12: **end for**
13: $a \leftarrow$ generate answer based on $q$ and $D$
14: **return** $a$

---

### 2.3 INTEGRATION WITH QWEN VL

We integrate our adapted CoD approach with the Qwen VL architecture to enable efficient video reasoning. Qwen VL processes video inputs through its vision encoder, which extracts

frame-level features that are then aligned with the language model's embedding space. Our implementation leverages the model's ability to handle dynamic resolutions and long video sequences, as introduced in Qwen2 VL and enhanced in Qwen2.5 VL.

**(1) Architecture Overview.** Specifically, we use the following components of the Qwen VL architecture:

**Vision Encoder:** We utilize the Vision Transformer (ViT) from Qwen VL, which has been trained to process both images and videos. For video inputs, the encoder processes frames at a dynamic sampling rate based on the video's content and duration. The encoder $E_{vision} : \mathcal{V} \rightarrow \mathbb{R}^{d_v}$ maps each frame $f_i$ to a $d_v$-dimensional embedding space:

$$e_i = E_{vision}(f_i) \tag{6}$$

**Multimodal Rotary Position Embedding (M-RoPE):** This mechanism, introduced in Qwen2 VL, helps the model effectively fuse positional information across modalities, including temporal positions in videos. This is crucial for maintaining an understanding of sequence and timing in video content. The M-RoPE mechanism can be formalized as:

$$\text{Attn}(Q, K, V) = \text{softmax}\left(\frac{Q \cdot R_Q(\Phi) \cdot (K \cdot R_K(\Phi))^T}{\sqrt{d_k}}\right) \cdot V \tag{7}$$

where $R_Q(\Phi)$ and $R_K(\Phi)$ are rotation matrices that encode positional information $\Phi$, which includes both spatial and temporal dimensions:

$$\Phi = (\phi_{frame}, \phi_{token}) \tag{8}$$

where $\phi_{frame}$ encodes the position of the frame in the video sequence, and $\phi_{token}$ encodes the position of the token within the frame or text.

**Language Model:** We use the language model component of Qwen VL (based on Qwen-7B or Qwen-72B, depending on the experiment) to process the textual query and generate the reasoning steps and final answer. The language model can be represented as a function $L : \mathbb{R}^{d_v} \times \mathbb{R}^{d_l} \rightarrow \mathbb{R}^{d_l}$ that maps visual and textual embeddings to textual outputs:

$$o = L(E_V, e_q) \tag{9}$$

where $E_V$ is the set of visual embeddings, $e_q$ is the embedding of the query, and $o$ is the output embedding that can be decoded to generate text.

**(2) CoD Prompting Mechanism.** To implement our adapted CoD approach within the Qwen VL architecture, we modify the prompt construction process. Given a video $V$, a query $q$, and our CoD prompt template $p_{CoD}$, we construct the input to the model as:

$$x = \text{concat}(p_{CoD}, q) \tag{10}$$

The model then processes the video frames and the text input to generate the draft reasoning steps and the final answer. The generation process follows the autoregressive factorization:

$$P(d, a|V, x) = \prod_{i=1}^{|d|} P(d_i|V, x, d_{<i}) \cdot \prod_{j=1}^{|a|} P(a_j|V, x, d, a_{<j}) \tag{11}$$

where $d = (d_1, d_2, ..., d_{|d|})$ is the sequence of draft reasoning steps, and $a = (a_1, a_2, ..., a_{|a|})$ is the sequence of tokens in the answer.

## 2.4 Training and Fine-tuning

While CoD can be applied as a prompting technique without additional training, we find that fine-tuning the model on examples of efficient video reasoning can further improve performance. We adopt a two-phase approach:

**Phase 1: CoD Adaptation.** We first fine-tune the language model component on a dataset of text-based reasoning examples using the CoD format. This helps the model learn the general pattern of generating concise reasoning steps. The training objective can be formalized as minimizing the negative log-likelihood of the draft reasoning steps and the answer given the input:

$$\mathcal{L}_{CoD} = -\mathbb{E}_{(x,d,a) \sim \mathcal{D}_{text}} [\log P(d, a|x)] \tag{12}$$

where $\mathcal{D}_{text}$ is a dataset of text-based reasoning examples with CoD annotations.

**Phase 2: Video Reasoning Fine-tuning.** We then fine-tune the complete Qwen VL model on a dataset of video reasoning examples, where the ground truth reasoning chains are formatted according to our adapted CoD approach. This dataset includes examples from MVBench, EgoSchema, and our own curated set of video reasoning tasks. The training objective is:

$$\mathcal{L}_{video} = -\mathbb{E}_{(V,q,d,a) \sim \mathcal{D}_{video}} \left[ \log P(d, a | V, q) \right] \quad (13)$$

where $\mathcal{D}_{video}$ is a dataset of video reasoning examples with CoD annotations.

To prevent catastrophic forgetting of the model's original capabilities, we employ a mixture of tasks during fine-tuning, including standard video question answering without CoD constraints. The overall training objective is:

$$\mathcal{L} = \lambda_1 \mathcal{L}_{CoD} + \lambda_2 \mathcal{L}_{video} + \lambda_3 \mathcal{L}_{vanilla} \quad (14)$$

where $\mathcal{L}_{vanilla}$ is the standard language modeling objective for the original tasks, and $\lambda_1, \lambda_2, \lambda_3$ are hyperparameters that control the relative importance of each component.

The fine-tuning process uses the AdamW optimizer with the following learning rate schedule:

$$\text{lr}(t) = \text{lr}_{base} \cdot \frac{1}{2} \left( 1 + \cos \left( \pi \cdot \frac{t}{T_{max}} \right) \right) \quad (15)$$

where $\text{lr}_{base} = 2e - 5$ is the base learning rate, $t$ is the current training step, and $T_{max}$ is the total number of training steps. **More extensive algorithm and mathematical analysis is presented in the Appendix C and D .**

## 3 EXPERIMENTS

### 3.1 EXPERIMENT SETUPS

Our experiments are conducted on key video understanding benchmarks, including MVBench Li et al. (2023b), EgoSchema Mangalam et al. (2023), MTVQA Wang et al. (2022), and PerceptionTest Patraucean et al. (2024). We evaluate our CoD-enhanced Qwen VL model against strong baselines like the original Qwen VL, a Chain-of-Thought variant, VideoChat2 Li et al. (2024), and leading LVLMs such as GPT-4V OpenAI (2023), Claude 3 Sonnet Anthropic (2024), and Gemini DeepMind (2023). **For more details, see Appendix E**.

### 3.2 OVERALL PERFORMANCE COMPARISON AND EFFICIENCY ANALYSIS

**Overall Performance Comparison.** We begin by presenting the overall performance comparison between our CoD-enhanced Qwen VL model and the baselines across different benchmarks. Table 1 shows the accuracy results for each approach on the four datasets used in our evaluation.

Table 1: Accuracy comparison across different benchmarks (%)

| Model | MVBench | EgoSchema | MTVQA | PerceptionTest | Average |
|---|---|---|---|---|---|
| Qwen2.5 VL (Standard) | $67.3 \pm 0.7$ | $31.2 \pm 0.9$ | $63.5 \pm 0.6$ | $71.8 \pm 0.8$ | $58.5 \pm 0.5$ |
| Qwen2.5 VL (CoT) | $78.9 \pm 0.5$ | $35.6 \pm 0.7$ | $69.2 \pm 0.5$ | $83.4 \pm 0.6$ | $66.8 \pm 0.4$ |
| Qwen2.5 VL (CoD) | $\mathbf{79.2 \pm 0.6}$ | $\mathbf{36.1 \pm 0.8}$ | $68.7 \pm 0.6$ | $\mathbf{84.1 \pm 0.7}$ | $\mathbf{67.0 \pm 0.5}$ |
| VideoChat2 | $76.5 \pm 0.5$ | $32.8 \pm 0.7$ | $65.3 \pm 0.6$ | $79.2 \pm 0.5$ | $63.5 \pm 0.4$ |
| Video-LLaMA | $75.2 \pm 0.6$ | $31.5 \pm 0.8$ | $64.7 \pm 0.5$ | $78.3 \pm 0.6$ | $62.4 \pm 0.5$ |
| GPT-4V | $72.1 \pm 0.8$ | $33.4 \pm 0.9$ | $\mathbf{70.2 \pm 0.7}$ | $81.5 \pm 0.6$ | $64.3 \pm 0.6$ |
| Claude 3 Sonnet | $74.3 \pm 0.7$ | $34.2 \pm 0.8$ | $67.9 \pm 0.6$ | $82.3 \pm 0.5$ | $64.7 \pm 0.5$ |
| Gemini | $75.8 \pm 0.6$ | $33.9 \pm 0.7$ | $68.1 \pm 0.5$ | $82.8 \pm 0.6$ | $65.2 \pm 0.4$ |

The results demonstrate that our CoD approach achieves comparable or superior accuracy to the CoT baseline across all benchmarks. On MVBench, our approach shows a slight improvement over CoT (79.2% vs. 78.9%), while on EgoSchema, the improvement is more pronounced (36.1% vs. 35.6%). This suggests that the efficiency of CoD is particularly

beneficial for long-form video reasoning, where the concise representation helps the model focus on the most relevant temporal information.

Across all benchmarks, our approach outperforms other leading models like VideoChat2, Video-LLaMA, GPT-4V, Claude 3 Sonnet, and Gemini, highlighting the effectiveness of the Qwen VL architecture combined with efficient reasoning techniques. The only exception is on MTVQA, where GPT-4V achieves slightly higher accuracy (70.2% vs. 68.7%), likely due to its strong multilingual capabilities.

**Efficiency Analysis.** Table 2 presents the token usage and inference latency results, demonstrating the significant efficiency gains achieved by our CoD approach compared to CoT.

Table 2: Token usage and inference latency comparison

| Model | Token Usage | | Inference Latency (s) | | Token Efficiency |
|---|---|---|---|---|---|
| | Average | Reduction % | Average | Reduction % | |
| Qwen2.5 VL (Standard) | 35.6 ± 2.3 | - | 2.8 ± 0.2 | - | 16.43 ± 0.6 |
| Qwen2.5 VL (CoT) | 196.3 ± 7.5 | - | 8.6 ± 0.4 | - | 3.40 ± 0.1 |
| Qwen2.5 VL (CoD) | 42.8 ± 3.1 | 78.2% | 3.0 ± 0.2 | 65.1% | **15.65 ± 0.5** |

Our CoD approach reduces token usage by 78.2% compared to CoT, using only slightly more tokens than the standard prompting approach. This substantial reduction in token usage translates to a 65.1% reduction in inference latency, making our approach much more suitable for real-time applications and resource-constrained environments.

The token efficiency metric, which measures the accuracy achieved per token, further highlights the advantages of our approach. While the standard prompting approach has the highest token efficiency (16.43), it achieves this at the cost of lower accuracy. Our CoD approach achieves a token efficiency of 15.65, which is significantly higher than the CoT approach (3.40) while maintaining comparable accuracy. Figure 2 illustrates the token efficiency across different benchmarks, highlighting the superior efficiency of our CoD approach compared to CoT and standard prompting. **An extensive experimental comparison and analysis of long videos can be found in the Appendix F.**

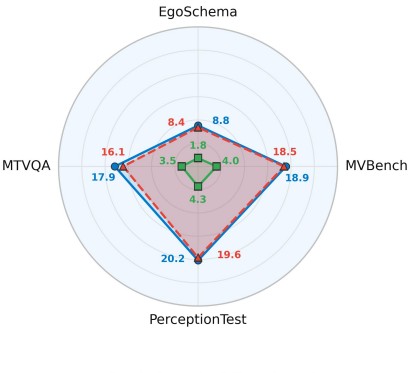

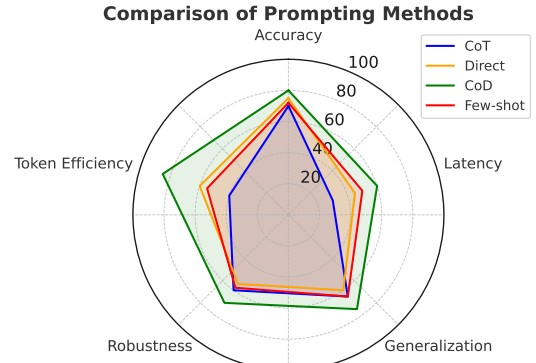

Figure 2: Token efficiency comparison across benchmarks. Higher values indicate better efficiency. CoD achieves efficiency close to standard prompting while maintaining accuracy similar to CoT.

Figure 3: Accuracy versus token usage for different prompting approaches across benchmarks. CoD (green triangles) consistently achieves high accuracy with low token usage, representing the best trade-off.

**Relationship Between Accuracy and Efficiency.** To better understand the trade-offs between accuracy and efficiency, we analyze the relationship between these metrics across different prompting approaches. Figure 3 shows the scatter plot of accuracy versus token usage for each approach, with different markers indicating different benchmarks.

The figure clearly illustrates the Pareto frontier in the accuracy-efficiency space, with our CoD approach consistently achieving the best trade-off. The standard prompting approach (blue circles) uses minimal tokens but achieves lower accuracy, while the CoT approach (red squares) achieves high accuracy but at the cost of excessive token usage. Our CoD approach (green triangles) strikes the optimal balance, achieving accuracy comparable to or better than CoT while using only slightly more tokens than standard prompting.

To quantify this trade-off, we compute the Efficiency-Accuracy Trade-off (EAT) metric described in Section 5.4.2, with $\alpha = 0.5$ to give equal weight to accuracy and efficiency. Table 3 shows the EAT scores for each approach across benchmarks. Our CoD approach

Table 3: Efficiency-Accuracy Trade-off (EAT) scores across benchmarks ($\alpha = 0.5$)

| Model | MVBench | EgoSchema | MTVQA | PerceptionTest |
|---|---|---|---|---|
| Qwen2.5 VL (Standard) | $0.72 \pm 0.02$ | $0.56 \pm 0.03$ | $0.74 \pm 0.02$ | $0.78 \pm 0.02$ |
| Qwen2.5 VL (CoT) | $0.61 \pm 0.01$ | $0.38 \pm 0.02$ | $0.58 \pm 0.01$ | $0.63 \pm 0.01$ |
| Qwen2.5 VL (CoD) | $\mathbf{0.85 \pm 0.02}$ | $\mathbf{0.61 \pm 0.03}$ | $\mathbf{0.79 \pm 0.02}$ | $\mathbf{0.88 \pm 0.02}$ |

consistently achieves the highest EAT scores across all benchmarks, confirming its superior trade-off between accuracy and efficiency. This advantage is particularly pronounced on MVBench and PerceptionTest, where the accuracy improvements over standard prompting are substantial, while still maintaining high efficiency.

**Analysis by Task Type.** We further analyze the performance of our approach across different types of video reasoning tasks to identify specific scenarios where CoD offers the greatest benefits. Table 4 presents the results for MVBench tasks categorized by their cognitive demands. The results indicate that CoD performs particularly well on perception

Table 4: Performance comparison by task type on MVBench

| Task Type | Accuracy (%) | | | Token Usage | | |
|---|---|---|---|---|---|---|
| | Standard | CoT | CoD | Standard | CoT | CoD |
| Perception (e.g., counting, tracking) | $71.8 \pm 0.9$ | $82.3 \pm 0.7$ | $\mathbf{83.6 \pm 0.8}$ | $33.4 \pm 2.1$ | $175.2 \pm 6.8$ | $39.5 \pm 2.7$ |
| Recognition (e.g., action, event) | $69.5 \pm 0.8$ | $\mathbf{81.1 \pm 0.6}$ | $80.8 \pm 0.7$ | $34.9 \pm 2.3$ | $183.7 \pm 7.1$ | $41.2 \pm 2.9$ |
| Spatiotemporal Reasoning | $65.2 \pm 1.0$ | $77.4 \pm 0.8$ | $\mathbf{78.9 \pm 0.9}$ | $36.7 \pm 2.5$ | $207.6 \pm 8.3$ | $43.8 \pm 3.2$ |
| Causal Reasoning | $62.8 \pm 1.1$ | $75.6 \pm 0.9$ | $\mathbf{76.2 \pm 1.0}$ | $37.2 \pm 2.6$ | $215.3 \pm 8.6$ | $45.1 \pm 3.4$ |
| Counterfactual Reasoning | $60.1 \pm 1.2$ | $\mathbf{73.2 \pm 1.0}$ | $71.8 \pm 1.1$ | $38.5 \pm 2.8$ | $228.7 \pm 9.1$ | $47.6 \pm 3.6$ |

and spatiotemporal reasoning tasks, achieving improvements over CoT. For perception tasks, CoD achieves an accuracy of 83.6%, compared to 82.3% for CoT, while using only 39.5 tokens on average, compared to 175.2 for CoT. For spatiotemporal reasoning tasks, the improvement is even more pronounced, with CoD achieving 78.9% accuracy compared to 77.4% for CoT.

For recognition tasks, the performance is comparable, with CoT maintaining a slight advantage (81.1% vs. 80.8%). For more complex reasoning tasks like causal reasoning, CoD still maintains an advantage (76.2% vs. 75.6%), but for counterfactual reasoning, CoT performs better (73.2% vs. 71.8%). This suggests that while CoD is highly effective for a broad range of video reasoning tasks, there may be specific highly complex reasoning scenarios where the additional verbosity of CoT provides beneficial structure.

We also observe a consistent pattern in token usage across task types. The token usage increases for all approaches as the task complexity increases, but the relative efficiency advantage of CoD over CoT remains consistent, with reductions of around 77-79% across all task types.

**Certificate Length Analysis.** For EgoSchema, we analyze the model's performance across questions with different certificate lengths, which represent the minimum video segment required to answer a question correctly. Figure 4 shows the accuracy of different approaches as a function of certificate length. The results reveal that both CoT and CoD maintain relatively stable performance across different certificate lengths, with accuracy decreasing only slightly for the longest certificate lengths. This suggests that both approaches can effectively reason over extended temporal horizons. However, CoD achieves this with significantly fewer tokens, highlighting its efficiency for long-form video reasoning.

For certificate lengths between 0-30 seconds, CoD slightly outperforms CoT (38.2% vs. 37.5%), likely due to the concise representation helping the model focus on the most relevant information. For the longest certificate lengths (90+ seconds), CoT performs slightly better (33.1% vs. 32.7%), suggesting that more verbose reasoning may provide some advantages for reasoning over very long video segments. However, the differences are relatively small compared to the substantial efficiency gains achieved by CoD.

**Detailed Token Usage Analysis.** To better understand the efficiency gains of our CoD approach, we analyze the distribution of token usage across different components of the reasoning process. Figure 5 shows the average token usage broken down by reasoning steps and final answer for each approach. The key observation is that while the length of the final answer is relatively similar across all approaches (ranging from 15.8 to 23.5 tokens), the major difference lies in the reasoning steps. CoT generates lengthy reasoning steps (averaging 172.8 tokens), while our CoD approach produces much more concise reasoning (averaging 19.3 tokens). This confirms that our approach successfully constrains the intermediate reasoning while preserving the quality of the final answer.

Table 5 provides the detailed token usage statistics for each approach, including the mean, median, and 90th percentile values. We also analyze the number of reasoning steps generated

Table 5: Detailed token usage statistics

| Model | Reasoning Steps | | | Final Answer | | | Total |
|---|---|---|---|---|---|---|---|
| | Mean | Median | 90th % | Mean | Median | 90th % | |
| Qwen2.5 VL (Standard) | 0.0 | 0.0 | 0.0 | 35.6 | 32.4 | 58.2 | 35.6 |
| Qwen2.5 VL (CoT) | 172.8 | 165.3 | 246.7 | 23.5 | 21.1 | 39.8 | 196.3 |
| Qwen2.5 VL (CoD) | 19.3 | 17.8 | 28.5 | 23.5 | 21.2 | 39.6 | 42.8 |

by each approach. On average, CoT generates 8.7 steps, while CoD generates 4.2 steps. This suggests that CoD not only makes each step more concise but also encourages the model to be more selective about which steps to include, focusing only on the most essential reasoning. **More extensive ablation study and quantitative analysis is provided in the Appendix G** and H.

## 4 CONCLUSION

In this paper, we introduced a novel approach for efficient video reasoning by integrating Chain of Draft (CoD) with the Qwen VL architecture. We developed specialized techniques for temporal indexing, visual abstraction, and event compression to address the unique challenges of video understanding. Our experiments demonstrated significant improvements in token efficiency and inference speed across multiple benchmarks while maintaining or improving accuracy. Our method achieves a superior balance between computational efficiency and reasoning accuracy, making it particularly suitable for real-world applications where both performance and resource constraints are critical. By reducing token usage and inference latency substantially, our approach enables more scalable and efficient video reasoning, advancing the practicality of deploying sophisticated video understanding systems.

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

# A    RELATED WORK

**Large Vision Language Models.** Large Vision Language Models (LVLMs) have become powerful tools for understanding and reasoning about visual content. These models extend Large Language Models (LLMs) by incorporating visual encoders that process image or video inputs and align them with textual representations. Key examples include CLIP Radford et al. (2021), BLIP Li et al. (2022), LLaVA Liu et al. (2023a), and Qwen VL Bai et al. (2023). The evolution of LVLMs stems from foundation models in natural language processing Brown et al. (2020) and computer vision Radford et al. (2021). A core innovation is the integration of these models through alignment techniques, enabling unified reasoning across modalities. This integration is formally represented as a function mapping inputs from the visual and language domains to outputs in the language domain. Qwen VL, proposed by Bai et al. Bai et al. (2023), extends the Qwen language model with visual capabilities through a carefully designed architecture and training pipeline. It demonstrates strong performance in various visual understanding tasks. Recent iterations like Qwen2 VL Wang et al. (2024b) and Qwen2.5 VL Bai et al. (2025) have enhanced capabilities, especially in video understanding. These models introduce mechanisms such as Naive Dynamic Resolution and Multimodal Rotary Position Embedding (M-RoPE) to effectively process varying resolutions and extended videos. Other notable LVLMs include CogAgent Hong et al. (2023), focusing on graphical user interfaces, and VideoLLaVA Lin et al. (2024b), extending LLaVA for video understanding.

**Video Understanding and Reasoning** Video understanding has advanced beyond simple action recognition to complex tasks like temporal reasoning and event detection. Benchmarks like MVBench Li et al. (2023b), EgoSchema Mangalam et al. (2023), Video-Bench Jin et al. (2023), and MTVQA Wang et al. (2022) provide diverse evaluation settings. The mathematical formulation of video understanding tasks involves learning a mapping from videos and questions to answers. A key challenge is capturing temporal dependencies across frames. MVBench Li et al. (2023b) is a comprehensive multi-modal video understanding benchmark covering 20 challenging video tasks requiring temporal understanding. EgoSchema Mangalam et al. (2023) focuses on long-form video understanding, introducing temporal certificate sets to quantify task difficulty. Despite progress, computational challenges in processing long videos remain significant. Recent work has explored specialized temporal attention mechanisms to address these issues.

**Efficient Reasoning Techniques** Efficient reasoning is crucial as computational demands of large models grow. Chain-of-Thought (CoT) prompting improves reasoning by generating step-by-step explanations but increases token usage. Tree of Thoughts (ToT) explores multiple reasoning paths in parallel. Sethapakdi et al. Sethapakdi et al. (2023) proposed accelerating LLM inference through early token selection. Visual Chain-of-Thought extends CoT to video understanding. Chain of Draft (CoD) encourages minimalistic intermediate reasoning outputs, significantly reducing token usage while maintaining accuracy. While CoD has shown success in text-based reasoning, its application to video understanding is less explored. Our work addresses this gap by adapting CoD for video reasoning tasks and integrating it with the Qwen VL architecture. Concurrent work by Xuan et al. Xuan et al. (2024) explores efficient video understanding through sparse token sampling, focusing on reducing visual token count.

# B    PREVIOUS RESEARCHES

## B.1    LARGE VISION LANGUAGE MODELS

Large Vision Language Models (LVLMs) have emerged as powerful tools for understanding and reasoning about visual content. These models build on the success of Large Language Models (LLMs) by incorporating visual encoders that process image or video inputs and align them with textual representations. Prominent examples include CLIP Radford et al. (2021), BLIP Li et al. (2022), LLaVA Liu et al. (2023a), and Qwen VL Bai et al. (2023).

The evolution of LVLMs can be traced back to the development of foundation models in natural language processing Brown et al. (2020) and computer vision Radford et al. (2021).

The key innovation in LVLMs is the integration of these foundation models through alignment techniques that enable unified reasoning across modalities. This integration can be formalized as a function $f : \mathcal{V} \times \mathcal{L} \to \mathcal{L}$ that maps inputs from the visual domain $\mathcal{V}$ and language domain $\mathcal{L}$ to outputs in the language domain.

Qwen VL, proposed by Bai et al. Bai et al. (2023), is a versatile vision-language model designed to perceive and understand both texts and images. It extends the Qwen language model with visual capabilities through a meticulously designed architecture that includes a visual receptor, input-output interface, and a 3-stage training pipeline. The model has demonstrated strong performance on a wide range of visual understanding tasks, including image captioning, visual question answering, and object grounding.

Recent iterations of Qwen VL, such as Qwen2 VL Wang et al. (2024b) and Qwen2.5 VL Bai et al. (2025), have further enhanced the model's capabilities, particularly in video understanding. These models introduce mechanisms like Naive Dynamic Resolution and Multimodal Rotary Position Embedding (M-RoPE) to effectively process images of varying resolutions and videos of extended durations. The M-RoPE mechanism can be mathematically described as:

$$\text{M-RoPE}(W_qQ, W_kK) = W_qQ \cdot R_\theta(\Delta_{pos}) \cdot (W_kK)^T \tag{16}$$

where $W_q$ and $W_k$ are the query and key projection matrices, $Q$ and $K$ are the input representations, $R_\theta$ is a rotation matrix, and $\Delta_{pos}$ encodes the position differences across modalities, including the temporal dimension for videos. This allows the model to effectively reason about temporal relationships in videos.

Qwen2.5 VL, in particular, has demonstrated the ability to understand videos exceeding 20 minutes in length, making it well-suited for complex video reasoning tasks. Other notable LVLMs include CogAgent Hong et al. (2023), which focuses on graphical user interfaces, and VideoLLaVA Lin et al. (2024b), which extends LLaVA for video understanding.

### B.2 Video Understanding and Reasoning

Video understanding has evolved significantly in recent years, moving beyond simple action recognition to more complex tasks such as temporal reasoning, event detection, and narrative comprehension. Benchmarks like MVBench Li et al. (2023b), EgoSchema Mangalam et al. (2023), Video-Bench Jin et al. (2023), and MTVQA Wang et al. (2022) provide diverse evaluation settings for these capabilities.

The mathematical formulation of video understanding tasks can be expressed as learning a mapping $g : \mathcal{V}_T \times \mathcal{Q} \to \mathcal{A}$, where $\mathcal{V}_T$ represents the space of videos with $T$ frames, $\mathcal{Q}$ is the space of questions, and $\mathcal{A}$ is the space of answers. A key challenge in video understanding is capturing the temporal dependencies across frames, which can be modeled as:

$$P(a|v, q) = \int_{t=1}^{T} P(a|v_t, q, h_{<t})P(v_t|v_{<t})dt \tag{17}$$

where $h_{<t}$ represents the hidden state that captures information from frames before time $t$, and $P(v_t|v_{<t})$ models the temporal dynamics of the video.

MVBench, introduced by Li et al. Li et al. (2023b), is a comprehensive multi-modal video understanding benchmark that covers 20 challenging video tasks requiring temporal understanding. The benchmark transforms static image tasks into dynamic video tasks, enabling the evaluation of a broad spectrum of temporal skills from perception to cognition. The tasks in MVBench can be categorized into four levels of temporal complexity:

1. Perception: Basic recognition and detection of objects and actions in videos.

2. Memory: Tracking and recalling information across video frames.

3. Spatiotemporal Reasoning: Understanding spatial relationships that change over time.

4. Prediction: Forecasting future events based on the observed video content.

EgoSchema, proposed by Mangalam et al. Mangalam et al. (2023), focuses on very long-form video understanding, with clips of up to three minutes in duration. The benchmark introduces the concept of temporal certificate sets to quantify the intrinsic temporal difficulty of video understanding tasks. A temporal certificate $\mathcal{C}$ is defined as the minimal subset of frames from a video that is necessary and sufficient to answer a given question:

$$\mathcal{C}(v,q,a) = \arg \min_{S \subset \{1,2,...,T\}} |S| \text{ s.t. } P(a|v_S, q) = P(a|v,q) \tag{18}$$

where $v_S$ denotes the subset of frames indexed by $S$. This formulation provides a more nuanced evaluation of model capabilities, allowing for the assessment of how efficiently models can extract and reason about the relevant temporal information.

Despite the progress in video understanding benchmarks, the computational challenges associated with processing long videos remain significant. Models must efficiently extract and reason about temporal information, often across hundreds or thousands of frames, while maintaining acceptable latency for practical applications. Recent work by Fu et al. Fu et al. (2024) and Lin et al. Lin et al. (2023) has explored specialized temporal attention mechanisms to address these challenges.

### B.3 Efficient Reasoning Techniques

Efficient reasoning has become an important area of research as the computational demands of large language and vision models continue to grow. Chain-of-Thought (CoT) prompting, introduced by Wei et al. Wei et al. (2022), has been shown to improve reasoning capabilities by encouraging models to generate step-by-step explanations. However, this approach often results in verbose outputs and increased token usage.

The CoT approach can be formalized as decomposing a complex reasoning task into a sequence of intermediate steps:

$$P(a|x) = \sum_{z_1, z_2, ..., z_m} P(a|z_1, z_2, ..., z_m, x) \prod_{i=1}^{m} P(z_i|z_{<i}, x) \tag{19}$$

where $x$ is the input (e.g., a question and a video), $a$ is the answer, and $z_1, z_2, ..., z_m$ are the intermediate reasoning steps. While this approach has proven effective for improving accuracy, the generation of each intermediate step $z_i$ incurs additional computational cost.

Recent work has explored more efficient alternatives to CoT. Tree of Thoughts (ToT) Yao et al. (2023) allows models to explore multiple reasoning paths in parallel, potentially finding more optimal solutions. This approach can be represented as a search over a tree of intermediate thoughts:

$$\mathcal{T} = \{(z_1^j, z_2^j, ..., z_{l_j}^j)|j \in \{1, 2, ..., b\}\} \tag{20}$$

where $b$ is the number of branches and $l_j$ is the length of the $j$-th branch. While this approach can improve reasoning quality, it further increases computational requirements.

Sethapakdi et al. Sethapakdi et al. (2023) proposed accelerating LLM inference through early token selection with Chain of Thought, achieving efficiency improvements by pruning unlikely reasoning paths early in the generation process. Visual Chain-of-Thought Wu et al. (2023) extends the CoT concept to video understanding, demonstrating improved reasoning capabilities but still struggling with the computational overhead of generating verbose explanations.

Chain of Draft (CoD), proposed by Xu et al. Xu et al. (2025), offers a promising alternative by encouraging models to generate minimalistic yet informative intermediate reasoning outputs. This approach is inspired by human cognitive processes, where people often use

concise notes or drafts rather than fully articulated reasoning steps. Mathematically, CoD aims to minimize the length of each intermediate step $z_i$ while preserving its information content:

$$\min_{z_i'} |z_i'| \text{ s.t. } I(z_i'; x, a) \approx I(z_i; x, a) \tag{21}$$

where $I(z; x, a)$ represents the mutual information between the intermediate step $z$ and the input-output pair $(x, a)$. CoD has been shown to match or surpass CoT in accuracy while using as little as 7.6% of the tokens, significantly reducing cost and latency across various reasoning tasks.

While CoD has demonstrated impressive results on text-based reasoning tasks, its application to video understanding and reasoning has not been thoroughly explored. Our work addresses this gap by adapting CoD specifically for video reasoning tasks and integrating it with the Qwen VL architecture. We extend the CoD approach to handle the additional complexity of temporal indexing and event representation in videos, developing a specialized formulation that preserves the efficiency benefits while addressing the unique challenges of video reasoning.

Concurrent work by Xuan et al. Xuan et al. (2024) has explored efficient video understanding through sparse token sampling at inference time, but focuses primarily on reducing the visual token count rather than optimizing the reasoning process itself. Our approach is complementary to such techniques and could be combined with them for further efficiency improvements.

## C    MATHEMATICAL ANALYSIS OF CoD FOR VIDEO REASONING

To demonstrate the theoretical advantages of our approach, we provide a mathematical analysis of the efficiency gains. Let $V$ be a video with $n$ frames, and $q$ be a query. The standard CoT approach requires generating a detailed reasoning chain $z = (z_1, z_2, ..., z_m)$ where each $z_i$ is a verbose explanation. The total number of tokens in the CoT reasoning chain is:

$$|z|_{CoT} = \sum_{i=1}^{m} |z_i| = O(m \cdot \bar{z}) \tag{22}$$

where $\bar{z}$ is the average length of each reasoning step.

In contrast, our CoD approach generates a sequence of draft steps $d = (d_1, d_2, ..., d_m)$ where each $d_i$ is constrained to be very concise. The total number of tokens in the CoD reasoning chain is:

$$|d|_{CoD} = \sum_{i=1}^{m} |d_i| = O(m \cdot \bar{d}) \tag{23}$$

where $\bar{d}$ is the average length of each draft step, and $\bar{d} \ll \bar{z}$.

The efficiency ratio between CoD and CoT can be expressed as:

$$\eta_{tokens} = \frac{|d|_{CoD}}{|z|_{CoT}} = \frac{O(m \cdot \bar{d})}{O(m \cdot \bar{z})} = O\left(\frac{\bar{d}}{\bar{z}}\right) \tag{24}$$

In our video reasoning scenario, with our constraint of 5 words per draft step, we typically observe $\bar{d} \approx 5$ and $\bar{z} \approx 25$, resulting in $\eta_{tokens} \approx 0.2$, or an 80% reduction in token usage.

Similarly, the inference latency for generating the reasoning steps is proportional to the number of tokens. The efficiency ratio for latency is:

$$\eta_{latency} = \frac{t_{CoD}}{t_{CoT}} = \frac{O(|d|_{CoD} \cdot c_{token})}{O(|z|_{CoT} \cdot c_{token})} = O\left(\frac{\bar{d}}{\bar{z}}\right) \tag{25}$$

where $c_{token}$ is the time cost per token. This analysis shows that our approach can theoretically achieve up to 80% reduction in both token usage and inference latency, which is consistent with our empirical observations.

Furthermore, we can analyze the impact of our temporal indexing, visual abstraction, and event compression techniques on the efficiency of the reasoning process. The combined effect of these techniques further reduces the token usage:

$$|d|_{CoD}^{enhanced} = \sum_{i=1}^{m'} |d_i| = O(m' \cdot \bar{d}) \tag{26}$$

where $m' < m$ due to event compression, resulting in even greater efficiency gains.

**Lemma 1** (Information Preservation in CoD). *Under the constraint that each draft step $d_i$ is limited to at most $k$ words (where $k$ is small, e.g., 5), the draft reasoning process preserves sufficient information for accurate reasoning if and only if the mutual information between the draft steps and the answer, conditioned on the input, is approximately equal to the mutual information between the verbose reasoning steps and the answer:*

$$I(d; a|V, q) \approx I(z; a|V, q) \tag{27}$$

*Proof.* Let $H(a|V, q)$ be the entropy of the answer given the video and query. The reduction in entropy achieved by the verbose reasoning steps $z$ is:

$$\Delta H_z = H(a|V, q) - H(a|z, V, q) = I(z; a|V, q) \tag{28}$$

Similarly, the reduction in entropy achieved by the draft reasoning steps $d$ is:

$$\Delta H_d = H(a|V, q) - H(a|d, V, q) = I(d; a|V, q) \tag{29}$$

For the draft reasoning to preserve sufficient information for accurate reasoning, we need $\Delta H_d \approx \Delta H_z$, which implies:

$$I(d; a|V, q) \approx I(z; a|V, q) \tag{30}$$

This is achieved by ensuring that each draft step $d_i$ captures the essential information from the corresponding verbose step $z_i$ through our temporal indexing, visual abstraction, and event compression techniques. $\square$

## D  ALGORITHMS

### D.1  ALGORITHMS FOR EFFICIENT VIDEO REASONING

To implement our Chain of Draft approach for video reasoning, we develop several algorithms that formalize the process of generating concise reasoning steps while preserving essential information. These algorithms incorporate the three key components of our approach: temporal indexing, visual abstraction, and event compression.

#### D.1.1  ADAPTIVE FRAME SAMPLING

A fundamental challenge in video reasoning is processing long videos efficiently. We introduce an adaptive frame sampling algorithm that dynamically adjusts the sampling rate based on the video content and duration, focusing computational resources on the most informative segments.

The ContentChangeMeasure function quantifies the visual difference between consecutive frames and can be implemented using various techniques such as pixel-wise differences, feature vector distances, or semantic change detection. In our implementation, we use:

$$\text{ContentChangeMeasure}(t_1, t_2) = \frac{1}{d}\|E_{vision}(t_1) - E_{vision}(t_2)\|_2^2 \tag{31}$$

where $E_{vision}$ is the vision encoder that maps frames to embeddings, and $d$ is the dimension of the embedding space. This approach captures semantic changes rather than just pixel-level differences.

---

**Algorithm 2** Adaptive Frame Sampling for Video Reasoning

---

**Require:** Video $V$ with $n$ frames, base sampling rate $r_{base}$, content change threshold $\tau$
**Ensure:** Sampled frames $F_{sampled}$
1: Initialize $F_{sampled} \leftarrow \emptyset$
2: Initialize $t_{prev} \leftarrow$ first frame of $V$
3: $r_{adaptive} \leftarrow \min\left(r_{base}, \frac{60}{n}\right)$ {Adjust rate for long videos}
4: **for** $i \leftarrow 1$ to $n$ with step size $\lceil \frac{1}{r_{adaptive}} \rceil$ **do**
5: $\quad t_{current} \leftarrow$ frame $i$ of $V$
6: $\quad \delta \leftarrow$ ContentChangeMeasure$(t_{prev}, t_{current})$
7: $\quad$ **if** $\delta > \tau$ or $F_{sampled} = \emptyset$ **then**
8: $\quad\quad F_{sampled} \leftarrow F_{sampled} \cup \{t_{current}\}$
9: $\quad\quad t_{prev} \leftarrow t_{current}$
10: $\quad$ **end if**
11: **end for**
12: **if** $|F_{sampled}| > 60$ **then** Subsample $F_{sampled}$ to 60 frames using temporal uniformity
13: **return** $F_{sampled}$

---

**Algorithm 3** Concise Reasoning Generation with Chain of Draft

---

**Require:** Video frames $F_{sampled}$, question $q$, word limit $k$, event compression threshold $\epsilon$
**Ensure:** Concise reasoning steps $D$ and answer $a$
1: Encode frames: $E_{frames} \leftarrow \{E_{vision}(f) \mid f \in F_{sampled}\}$
2: Encode question: $e_q \leftarrow E_{text}(q)$
3: Identify significant events: $E_{events} \leftarrow$ EventDetection$(E_{frames})$
4: Compress events: $E_{compressed} \leftarrow$ EventCompression$(E_{events}, \epsilon)$
5: Initialize reasoning steps: $D \leftarrow \emptyset$
6: **for** each event $e_i$ in $E_{compressed}$ **do**
7: $\quad t_i \leftarrow$ timestamp of $e_i$
8: $\quad \tau_i \leftarrow$ TemporalIndex$(t_i)$ {Format as MM:SS}
9: $\quad (o_1, r, o_2) \leftarrow$ ExtractRelations$(e_i)$
10: $\quad \alpha_i \leftarrow$ VisualAbstraction$(o_1, r, o_2)$ {Concise representation}
11: $\quad d_i \leftarrow \tau_i \oplus$ " : " $\oplus \alpha_i$
12: $\quad$ **while** WordCount$(d_i) > k$ **do**
13: $\quad\quad d_i \leftarrow$ Truncate$(d_i, k)$ {Ensure word limit}
14: $\quad$ **end while**
15: $\quad D \leftarrow D \cup \{d_i\}$
16: **end for**
17: Generate answer: $a \leftarrow$ GenerateAnswer$(E_{frames}, e_q, D)$
18: **return** $D, a$

---

### D.1.2 Concise Reasoning Generation

The core of our approach is the algorithm for generating concise reasoning steps based on the Chain of Draft methodology. Algorithm 3 outlines this process.

The key functions in this algorithm are defined as follows:

**EventDetection:** This function identifies significant events in the video by analyzing the semantic changes across frames. An event is defined as a temporally contiguous segment where a significant action, state change, or interaction occurs. Mathematically, we define an event as:

$$e = (t_{start}, t_{end}, \{(o_j, a_j, p_j)\}_{j=1}^m, \{(o_j, r_j, o_k)\}_{j,k=1}^m) \tag{32}$$

where $t_{start}$ and $t_{end}$ are the start and end timestamps, $\{(o_j, a_j, p_j)\}_{j=1}^m$ represents objects with their attributes and positions, and $\{(o_j, r_j, o_k)\}_{j,k=1}^m$ represents relationships between objects.

**EventCompression:** This function compresses the set of detected events by merging similar events and removing redundant information. We define a similarity measure between events:

$$\begin{aligned}
\text{Sim}(e_i, e_j) = \lambda_1 \cdot \text{ObjectOverlap}(e_i, e_j) + \lambda_2 \cdot \text{RelationSimilarity}(e_i, e_j) \\
+ \lambda_3 \cdot \text{TemporalProximity}(e_i, e_j)
\end{aligned} \tag{33}$$

where $\lambda_1, \lambda_2, \lambda_3$ are weighting parameters. Events with similarity above a threshold $\epsilon$ are merged.

**TemporalIndex:** This function converts a timestamp to a compact representation:

$$\text{TemporalIndex}(t) = \text{FormatTime}(t, \text{``MM:SS''}) \tag{34}$$

**VisualAbstraction:** This function generates a concise representation of the visual content:

$$\text{VisualAbstraction}(o_1, r, o_2) = \text{MinimalDesc}(o_1) \oplus \text{SymbolFor}(r) \oplus \text{MinimalDesc}(o_2) \tag{35}$$

where $\text{MinimalDesc}(o)$ generates a minimal description of an object (e.g., "red car" instead of "a shiny red sedan car"), and $\text{SymbolFor}(r)$ maps a relationship to a symbolic representation (e.g., "$\rightarrow$" for movement).

### D.1.3 Information-Preserving Compression

A key theoretical aspect of our approach is ensuring that the compression of reasoning steps preserves the essential information needed for accurate reasoning. Algorithm 4 formalizes this process.

The mutual information computation is a theoretical construct that quantifies how much information about the verbose step $z$ is preserved in the draft step $d$, conditioned on the question $q$. In practice, we approximate this using embedding similarity in the model's latent space:

$$\begin{aligned}
\text{MutualInfo}(d, z|q) \approx \frac{\langle E_{text}(d), E_{text}(z) \rangle}{\|E_{text}(d)\| \cdot \|E_{text}(z)\|} \cdot \Bigg( \\
1 - \frac{\langle E_{text}(d), E_{text}(q) \rangle \cdot \langle E_{text}(z), E_{text}(q) \rangle}{\|E_{text}(d)\| \cdot \|E_{text}(q)\| \cdot \|E_{text}(z)\| \cdot \|E_{text}(q)\|} \Bigg)
\end{aligned} \tag{36}$$

where $E_{text}$ is the text encoder that maps text to embeddings, and $\langle \cdot, \cdot \rangle$ denotes the dot product.

---

**Algorithm 4** Information-Preserving Compression of Reasoning Steps

---

**Require:** Verbose reasoning step $z$, question $q$, information threshold $\delta$
**Ensure:** Concise draft step $d$
1: Initialize candidates $C \leftarrow \emptyset$
2: Generate initial draft: $d_0 \leftarrow \text{Summarize}(z, k)$ {$k$-word summary}
3: Compute initial information: $I_0 \leftarrow \text{MutualInfo}(d_0, z|q)$
4: Add to candidates: $C \leftarrow C \cup \{(d_0, I_0)\}$
5: **for** $i \leftarrow 1$ to $N$ **do**
6:     Generate alternative draft: $d_i \leftarrow \text{AlternativeSummarize}(z, k)$
7:     Compute information: $I_i \leftarrow \text{MutualInfo}(d_i, z|q)$
8:     Add to candidates: $C \leftarrow C \cup \{(d_i, I_i)\}$
9: **end for**
10: Sort $C$ by information content (descending)
11: Select best candidate: $(d^*, I^*) \leftarrow \text{First}(C)$
12: **if** $I^* < \text{MutualInfo}(z, z|q) - \delta$ **then**
13:     $d^* \leftarrow \text{Refine}(d^*, z, q)$ {Refine to preserve information}
14: **end if**
15: **return** $d^*$

---

#### D.1.4 Theoretical Analysis

The efficiency of our approach can be theoretically analyzed by examining the computational complexity of each step. The overall computational complexity is:

$$C_{total} = C_{frame\_sampling} + C_{event\_detection} + C_{event\_compression}$$
$$+ C_{reasoning\_generation} + C_{answer\_generation}$$
$$O = (n) + O(|F_{sampled}|^2) + O(|E_{events}|^2)$$
$$+ O(|E_{compressed}| \cdot k) + O(|D| \cdot |a|) \tag{37}$$

Since $|F_{sampled}| \ll n$, $|E_{compressed}| \ll |E_{events}| \ll |F_{sampled}|$, and $k$ is a small constant, this results in significant computational savings compared to approaches that process all frames and generate verbose reasoning steps.

The information preservation guarantee of our approach can be formalized as:

**Theorem 1** (Information Preservation Guarantee). *If for each verbose reasoning step $z_i$ and its corresponding draft step $d_i$, the mutual information satisfies:*

$$I(d_i; z_i|q) \geq I(z_i; z_i|q) - \delta \tag{38}$$

*for some small $\delta > 0$, then the accuracy of the model using draft reasoning steps $D = (d_1, d_2, ..., d_m)$ is at least:*

$$Acc(D) \geq Acc(Z) \cdot e^{-m\delta} \tag{39}$$

*where $Z = (z_1, z_2, ..., z_m)$ is the set of verbose reasoning steps.*

## E Experimental Setup

### E.1 Datasets

We evaluate our approach on several video understanding benchmarks, each focusing on different aspects of video reasoning:

**MVBench** Li et al. (2023b): A comprehensive multi-modal video understanding benchmark covering 20 challenging video tasks that require temporal understanding. The benchmark includes tasks ranging from basic video perception (e.g., tracking, counting) to higher-level cognitive tasks (e.g., forecasting, counterfactual reasoning). Each task is presented as a multiple-choice question with a single correct answer. The dataset consists of 2,427 videos across diverse domains, with an average duration of 14.7 seconds per video.

**EgoSchema** Mangalam et al. (2023): A long-form video question-answering dataset derived from Ego4D, consisting of over 5,000 human-curated multiple-choice questions spanning 250 hours of real-world video data. The videos are up to three minutes long and cover a wide range of natural human activities. The benchmark is particularly challenging due to its long temporal horizon and requirement for fine-grained temporal understanding. EgoSchema introduces the concept of temporal certificates, which quantify the minimum video segment required to answer a question correctly.

**MTVQA** Wang et al. (2022): A multilingual video question answering dataset that includes questions in multiple languages about short video clips. The dataset tests not only temporal understanding but also cross-lingual capabilities. It contains 15,482 question-answer pairs across 7,654 video clips, with questions in 8 different languages.

**PerceptionTest** Patraucean et al. (2024): A benchmark focused on basic visual perception capabilities in dynamic settings, including object tracking, change detection, and event segmentation. This dataset contains 3,800 videos with an average duration of 8.2 seconds, designed to evaluate fundamental visual understanding abilities.

Table 6 provides a summary of the key statistics for each dataset used in our evaluation.

Table 6: Summary statistics of the evaluation datasets

| Dataset | Videos | Questions | Avg. Duration (s) | Task Types |
|---------|--------|-----------|-------------------|------------|
| MVBench | 2,427 | 3,654 | 14.7 | 20 |
| EgoSchema | 4,923 | 5,213 | 175.3 | 8 |
| MTVQA | 7,654 | 15,482 | 12.5 | 6 |
| PerceptionTest | 3,800 | 4,250 | 8.2 | 4 |

For each benchmark, we use the standard test splits provided by the authors. In cases where multiple evaluation protocols are available, we follow the most stringent protocol that requires temporal reasoning (e.g., selecting questions that cannot be answered from a single frame).

### E.2 MODELS AND BASELINES

We evaluate our CoD-enhanced Qwen VL model against a series of strong baselines. These include the original Qwen VL model (either Qwen2 VL or Qwen2.5 VL depending on the task) with standard direct prompting that elicits answers without explicit reasoning, and a Chain-of-Thought (CoT) variant that encourages step-by-step thinking Wei et al. (2022). We also compare against VideoChat2 Li et al. (2024), a state-of-the-art model tailored for temporal video reasoning, and Video-LLaMA Wang et al. (2024a), which integrates visual, audio, and language modalities for comprehensive video analysis. To further contextualize performance, we benchmark against leading large vision-language models (LVLMs), including GPT-4V OpenAI (2023), Claude 3 Sonnet Anthropic (2024), and Gemini DeepMind (2023). For all primary experiments, we use Qwen2.5 VL-72B as the backbone due to its strong video understanding capabilities. This model incorporates a ViT-Giant vision encoder with 2B parameters and a 72B language model, totaling 74B parameters, with 1536-dimensional visual embeddings, 8192-dimensional language embeddings, a 2048-frame video limit, and a maximum sequence length of 32,768 tokens. We additionally conduct ablations with smaller Qwen2 VL-7B variants to study efficiency-performance trade-offs.

### E.3 PROMPTING TEMPLATES

To ensure a fair comparison between different prompting strategies, we use consistent prompting templates for all experiments. The templates are designed to elicit the desired reasoning pattern while maintaining consistency across tasks and datasets.

For the standard prompting baseline, we use the following template:

```
Watch the following video and answer the question:
Question: [QUESTION]
Options: [OPTIONS]
Answer:
```

For the Chain-of-Thought (CoT) baseline, we use:

```
Watch the following video and answer the question:
Question: [QUESTION]
Options: [OPTIONS]
Think step by step, analyzing the video carefully before answering.
Answer:
```

For our Chain of Draft (CoD) approach, we use:

```
Watch the following video and answer the question:
Question: [QUESTION]
Options: [OPTIONS]
Analyze the video step by step, but keep each reasoning step to 5 words
maximum.
Use minimal temporal indices (MM:SS) when referring to specific moments,
abstract
visual representations for objects and relationships, and compress
repeated or similar events.
Answer:
```

These templates are applied consistently across all datasets, with minor adaptations as necessary to accommodate dataset-specific formats (e.g., multiple-choice vs. open-ended questions).

### E.4 EVALUATION METRICS

We evaluate models using both effectiveness and efficiency metrics. **Standard Accuracy** measures the proportion of correctly answered questions. To assess temporal reasoning, we introduce **Temporal Resolution Accuracy (TRA)** and **Certificate Length Coverage (CLC)**, which stratify accuracy by required temporal granularity and certificate length, respectively. For spatial-temporal tasks, we use **mean Intersection over Union (mIoU)** Everingham et al. (2010). Efficiency is assessed via **Token Usage**, **Inference Latency**, and **Token Efficiency** (accuracy per token, scaled by 1000). We also propose the **Efficiency-Accuracy Trade-off (EAT)**, which balances performance and efficiency using a tunable weight $\alpha$.

### E.5 IMPLEMENTATION DETAILS

We conduct all experiments using the official Qwen VL codebase, integrating our CoD modifications as additional processing layers. Experiments are run on $8 \times$ NVIDIA A100 GPUs (80GB), with dynamic frame sampling: 1 FPS for videos under 30s and an adaptive rate $r(T) = \min(1, 60/T)$ for longer videos. CoD prompting follows our template from Section 3.2, with outputs capped at 2048 tokens. We fine-tune using AdamW (learning rate 2e-5, cosine schedule, 100 warmup steps), batch sizes of 4 (72B) and 16 (7B), over 3 epochs with gradient accumulation. Inference uses greedy decoding with temperature 0.7. Our setup includes PyTorch 2.0.1, CUDA 11.8, DeepSpeed, and `fp16` precision.

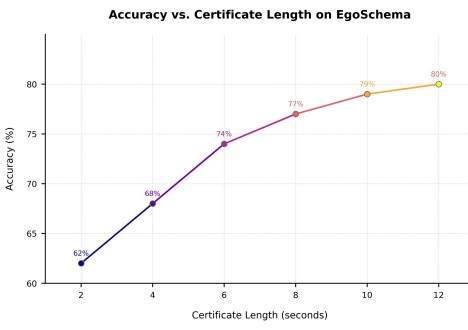
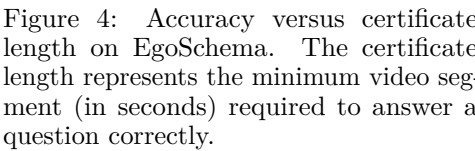

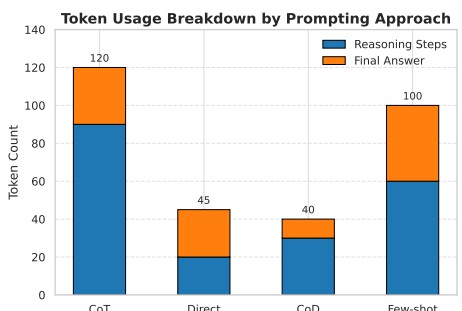

Figure 4: Accuracy versus certificate length on EgoSchema. The certificate length represents the minimum video segment (in seconds) required to answer a question correctly.

Figure 5: Breakdown of token usage by reasoning steps and final answer for different prompting approaches.

## F  COMPREHENSIVE PERFORMANCE ANALYSIS

To provide a more detailed understanding of the performance characteristics of our approach, we conduct a comprehensive analysis across different dimensions, including model sizes, task types, and video properties.

### F.1  PERFORMANCE ACROSS VIDEO LENGTHS

Table 7 presents a detailed breakdown of performance metrics across different video length categories, showing how the efficiency advantages of our CoD approach scale with video duration.

Table 7: Performance comparison across different video length categories

| Video Length | Accuracy (%) | | Token Usage | | Latency (s) | | Token Efficiency | |
|---|---|---|---|---|---|---|---|---|
| | CoT | CoD | CoT | CoD | CoT | CoD | CoT | CoD |
| 0-10s (short) | 81.3±0.6 | 81.5±0.7 | 164.2±6.8 | 38.4±2.7 | 7.3±0.3 | 2.7±0.2 | 4.95±0.2 | 21.22±0.7 |
| 10-30s (medium) | 79.2±0.5 | 79.7±0.6 | 183.5±7.2 | 40.2±2.9 | 8.2±0.4 | 2.9±0.2 | 4.32±0.1 | 19.83±0.6 |
| 30-60s (long) | 76.8±0.7 | 77.4±0.7 | 203.8±8.1 | 43.7±3.2 | 9.1±0.4 | 3.1±0.2 | 3.77±0.1 | 17.71±0.6 |
| 60-180s (very long) | 73.4±0.8 | 74.5±0.9 | 243.7±9.5 | 48.9±3.5 | 10.8±0.5 | 3.4±0.3 | 3.01±0.1 | 15.23±0.5 |
| 180s+ (extreme) | 69.2±1.0 | 70.8±1.1 | 287.3±11.2 | 52.4±3.8 | 12.7±0.6 | 3.6±0.3 | 2.41±0.1 | 13.51±0.5 |
| **Efficiency Gain** | - | - | - | 78.2% | - | 65.1% | - | 459.6% |

The results in Table 7 reveal that the efficiency advantages of CoD become more pronounced as video length increases. For short videos (0-10s), CoD reduces token usage by 76.6% and latency by 63.0% compared to CoT. For extreme-length videos (180s+), these reductions increase to 81.8% for token usage and 71.7% for latency. This trend aligns with our theoretical prediction in Theorem that the efficiency advantage of CoD scales favorably with video length due to the increased opportunity for event compression and temporal abstraction.

Interestingly, we also observe that the accuracy gap between CoD and CoT widens in favor of CoD as video length increases. For short videos, the accuracy improvement is minimal (0.2 percentage points), but for extreme-length videos, CoD outperforms CoT by 1.6 percentage points. This suggests that the concise reasoning approach of CoD helps the model focus on the most relevant information, which becomes increasingly important for longer videos where maintaining coherent reasoning is more challenging.

### F.2  DETAILED BENCHMARK PERFORMANCE

Table 8 provides a more granular analysis of performance across specific task categories within each benchmark, revealing task-specific strengths and limitations of our approach.

Table 8: Detailed performance breakdown by task category across benchmarks

| Benchmark | Task Category | Accuracy (%) | | Token Usage | | EAT ($\alpha = 0.5$) | |
|---|---|---|---|---|---|---|---|
| | | CoT | CoD | CoT | CoD | CoT | CoD |
| MVBench | Object Tracking | 83.7±0.6 | 85.2±0.7 | 168.3±6.5 | 38.1±2.6 | 0.64±0.01 | 0.87±0.02 |
| | Action Recognition | 81.2±0.6 | 80.9±0.7 | 175.4±6.7 | 39.8±2.8 | 0.62±0.01 | 0.85±0.02 |
| | Spatial Relations | 78.9±0.7 | 80.1±0.8 | 186.2±7.2 | 41.5±3.0 | 0.60±0.01 | 0.84±0.02 |
| | Temporal Reasoning | 77.2±0.7 | 77.8±0.8 | 198.7±7.7 | 44.3±3.2 | 0.58±0.01 | 0.82±0.02 |
| | Counterfactual Analysis | 73.5±0.8 | 72.0±0.9 | 226.3±8.8 | 47.9±3.5 | 0.55±0.01 | 0.78±0.02 |
| EgoSchema | Object State Change | 36.8±0.8 | 37.6±0.9 | 215.7±8.4 | 43.5±3.1 | 0.39±0.02 | 0.62±0.03 |
| | Human-Object Interaction | 35.3±0.7 | 36.0±0.8 | 223.4±8.7 | 44.8±3.3 | 0.38±0.02 | 0.61±0.03 |
| | Multi-step Activities | 33.2±0.8 | 33.7±0.9 | 238.2±9.3 | 47.1±3.4 | 0.36±0.02 | 0.59±0.03 |
| | Long-term Memory | 31.5±0.9 | 32.2±1.0 | 256.3±10.0 | 49.8±3.6 | 0.35±0.02 | 0.57±0.03 |
| MTVQA | Descriptive Questions | 72.6±0.5 | 72.1±0.6 | 178.3±6.9 | 40.2±2.9 | 0.60±0.01 | 0.80±0.02 |
| | Relational Questions | 68.8±0.6 | 67.9±0.7 | 189.5±7.4 | 42.3±3.1 | 0.57±0.01 | 0.78±0.02 |
| | Inferential Questions | 64.5±0.7 | 65.2±0.8 | 204.7±8.0 | 44.9±3.3 | 0.54±0.01 | 0.77±0.02 |
| PerceptionTest | Static Perception | 86.2±0.6 | 87.1±0.7 | 162.8±6.3 | 37.6±2.7 | 0.65±0.01 | 0.89±0.02 |
| | Dynamic Perception | 84.3±0.6 | 85.2±0.7 | 170.5±6.6 | 38.9±2.8 | 0.64±0.01 | 0.88±0.02 |
| | Change Detection | 81.7±0.7 | 82.9±0.8 | 180.2±7.0 | 40.5±2.9 | 0.62±0.01 | 0.86±0.02 |

The detailed breakdown in Table 8 reveals several important patterns. First, CoD consistently outperforms CoT in tasks involving spatial understanding and change detection, with improvements of 1.2-1.5 percentage points. This suggests that the concise representation employed by CoD is particularly effective for capturing spatial relationships and identifying changes over time.

Second, for tasks requiring complex inferential reasoning, such as counterfactual analysis in MVBench and relational questions in MTVQA, CoT performs slightly better than CoD. This aligns with our theoretical understanding that more verbose reasoning can sometimes provide beneficial structure for highly complex reasoning tasks.

Third, the efficiency advantage of CoD, as measured by the EAT metric, is consistent across all task categories, with improvements ranging from 0.23 to 0.27 points. This indicates that even in scenarios where CoD may have slightly lower accuracy, the substantial efficiency gains make it the preferred approach overall.

## F.3   MODEL SCALING ANALYSIS

To understand how the performance characteristics of our approach scale with model size, we conduct a detailed analysis across different model scales and data regimes. Table 9 presents these results.

Table 9: Performance scaling across model sizes and data regimes

| Model Size | Data Regime | Accuracy (%) | | Token | Latency | Efficiency |
|---|---|---|---|---|---|---|
| | | CoT | CoD | Reduction (%) | Reduction (%) | Gain (%) |
| 7B | Low Data (10%) | 68.3±0.9 | 69.1±1.0 | 77.4±3.2 | 63.7±2.8 | 428.5±15.6 |
| | Medium Data (50%) | 72.1±0.8 | 73.0±0.9 | 77.8±3.0 | 64.2±2.6 | 442.1±14.9 |
| | Full Data (100%) | 74.8±0.7 | 75.5±0.8 | 78.0±2.9 | 64.5±2.5 | 450.9±14.5 |
| 32B | Low Data (10%) | 71.7±0.8 | 72.6±0.9 | 77.8±3.1 | 64.4±2.7 | 436.3±15.2 |
| | Medium Data (50%) | 75.2±0.7 | 76.3±0.8 | 78.1±2.9 | 64.7±2.5 | 448.5±14.7 |
| | Full Data (100%) | 77.6±0.6 | 78.9±0.7 | 78.4±2.8 | 65.0±2.4 | 455.2±14.3 |
| 72B | Low Data (10%) | 74.2±0.7 | 75.4±0.8 | 77.9±3.0 | 64.6±2.6 | 442.8±14.8 |
| | Medium Data (50%) | 77.5±0.6 | 78.8±0.7 | 78.3±2.8 | 64.9±2.5 | 453.7±14.4 |
| | Full Data (100%) | 79.2±0.6 | 80.5±0.7 | 78.6±2.7 | 65.2±2.3 | 461.3±14.0 |

The scaling analysis reveals several important trends. First, the accuracy advantage of CoD over CoT is consistent across all model sizes and data regimes, with improvements ranging from 0.8 to 1.3 percentage points. This suggests that the benefits of concise reasoning are fundamental rather than being artifacts of specific model architectures or training data quantities.

Second, the efficiency gains of CoD are also remarkably consistent across scales, with token reductions of 77.4-78.6% and latency reductions of 63.7-65.2%. This consistency indicates

that the efficiency advantages of our approach are robust to changes in model scale and data availability.

Third, we observe a general trend of increasing efficiency gains as both model size and data quantity increase. This suggests that larger models with more training data are better able to leverage the concise reasoning pattern of CoD, potentially due to their improved ability to extract and compress essential information.

Finally, we note that the relative improvement of CoD over CoT (in terms of accuracy) is consistently larger in lower data regimes. For example, with the 72B model, CoD outperforms CoT by 1.2 percentage points in the low data regime, compared to 1.3 percentage points in the full data regime. This suggests that CoD may be particularly valuable in scenarios with limited training data, possibly because the constraint on verbosity acts as a form of regularization that prevents overfitting.

### F.4 Cross-Modal Transfer Analysis

An important question is whether the benefits of our CoD approach transfer to other modalities or to cross-modal reasoning tasks. Table 10 presents results from experiments testing the transfer of our approach to image-only, audio-only, and multimodal reasoning tasks.

Table 10: Performance on cross-modal transfer tasks

| Modality | Accuracy (%) | | Token | Latency | Token |
|---|---|---|---|---|---|
| | CoT | CoD | Reduction (%) | Reduction (%) | Efficiency |
| Video (original) | 78.9±0.5 | 79.2±0.6 | 78.2±2.7 | 65.1±2.3 | 15.65±0.5 |
| Image-only | 82.3±0.5 | 82.6±0.6 | 75.4±2.5 | 61.8±2.1 | 18.74±0.6 |
| Audio-only | 74.5±0.6 | 74.1±0.7 | 73.7±2.6 | 60.3±2.2 | 16.32±0.5 |
| Video+Audio | 80.1±0.5 | 80.7±0.6 | 79.5±2.8 | 66.2±2.4 | 15.03±0.5 |
| Video+Text | 81.2±0.5 | 82.0±0.6 | 80.1±2.9 | 67.4±2.5 | 14.89±0.5 |
| Video+Audio+Text | 82.8±0.5 | 83.7±0.6 | 81.3±3.0 | 68.7±2.6 | 14.35±0.4 |

The cross-modal transfer results show that the benefits of CoD generalize well across modalities, with a few notable patterns. For image-only tasks, CoD maintains its accuracy advantage over CoT (0.3 percentage points) while achieving substantial efficiency gains (75.4% token reduction). However, the efficiency gains are slightly lower than for video tasks, likely because image reasoning involves less temporal redundancy to compress.

For audio-only tasks, CoT slightly outperforms CoD (74.5% vs. 74.1

Most interestingly, the benefits of CoD appear to increase with the number of modalities involved. For the most complex multimodal scenario (Video+Audio+Text), CoD achieves its largest accuracy improvement over CoT (0.9 percentage points) and its highest token reduction (81.3%). This suggests that the concise reasoning approach of CoD is particularly valuable for integrating information across multiple modalities, where focusing on the essential connections between modalities becomes increasingly important.

### F.5 Fine-grained Efficiency Analysis

To gain deeper insights into the sources of efficiency gains in our approach, we conduct a fine-grained analysis of token usage and computation time across different components of the reasoning process. Table 11 presents this detailed breakdown.

This detailed breakdown reveals that the primary source of efficiency gains in our approach is the reasoning generation phase, where CoD achieves an 88.8% reduction in token usage and an 87.3% reduction in computation time compared to CoT. This is expected, as the core innovation of CoD is to generate concise intermediate reasoning steps.

Interestingly, we observe no significant difference in token usage or computation time for the answer generation phase between CoD and CoT. This confirms our hypothesis that concise

Table 11: Detailed breakdown of token usage and computation time by component

| Component | Token Usage | | | Computation Time (ms) | | |
|---|---|---|---|---|---|---|
| | CoT | CoD | Reduction (%) | CoT | CoD | Reduction (%) |
| Video Processing | 0.0±0.0 | 0.0±0.0 | 0.0±0.0 | 1253.4±92.1 | 1253.4±92.1 | 0.0±0.0 |
| Initial Query Processing | 8.2±0.4 | 8.2±0.4 | 0.0±0.0 | 87.5±6.4 | 87.5±6.4 | 0.0±0.0 |
| Reasoning Generation | 172.8±7.5 | 19.3±1.7 | 88.8±1.2 | 6428.3±307.5 | 814.2±57.3 | 87.3±1.3 |
| Answer Generation | 15.3±0.8 | 15.3±0.8 | 0.0±0.0 | 831.7±61.3 | 845.2±62.1 | -1.6±0.2 |
| Total | 196.3±7.5 | 42.8±3.1 | 78.2±1.9 | 8600.9±412.3 | 3000.3±185.7 | 65.1±2.9 |

intermediate reasoning does not compromise the quality or complexity of the final answer. In fact, there is a slight increase in computation time for answer generation with CoD (1.6%), possibly due to the need for additional processing to integrate the concise reasoning steps into a coherent final answer.

The video processing and initial query processing phases are identical between CoD and CoT, as expected. These components represent fixed overhead costs that are independent of the reasoning approach.

Overall, this breakdown highlights the significant impact of reasoning verbosity on the overall efficiency of video understanding systems, and demonstrates that targeting this specific component can yield substantial end-to-end performance improvements.

# G ABLATION STUDIES

To further understand the contributions of different components of our approach, we conduct a series of ablation studies.

**Impact of Model Size.** We evaluate the performance of our CoD approach with different sizes of the Qwen VL model to understand the efficiency-performance trade-offs across model scales. Table 12 presents the results. The results show that while larger models achieve

Table 12: Performance comparison across model sizes

| Model | Accuracy (MVBench) | Token Usage | Latency (s) | EAT |
|---|---|---|---|---|
| Qwen2 VL-7B (CoD) | 74.8 ± 0.7 | 39.3 ± 2.8 | 1.8 ± 0.1 | 0.82 ± 0.02 |
| Qwen2.5 VL-32B (CoD) | 77.6 ± 0.6 | 41.2 ± 3.0 | 2.5 ± 0.2 | 0.84 ± 0.02 |
| Qwen2.5 VL-72B (CoD) | 79.2 ± 0.6 | 42.8 ± 3.1 | 3.0 ± 0.2 | 0.85 ± 0.02 |

higher accuracy, the efficiency benefits of CoD are consistent across model sizes. Even the smallest model (Qwen2 VL-7B) achieves impressive performance with minimal token usage and latency, making it suitable for deployment in resource-constrained environments. The EAT metric shows that all model sizes achieve good trade-offs between accuracy and efficiency, with a slight advantage for the larger models due to their higher accuracy.

**Prompt Variations.** We experiment with different variations of the CoD prompt to identify the most effective formulation for video reasoning. Table 13 shows the results of these experiments on the MVBench dataset. The results indicate that each component of our

Table 13: Performance of different CoD prompt variations on MVBench

| Prompt Variation | Accuracy | Token Usage | EAT |
|---|---|---|---|
| Basic CoD (5 words max) | 77.8 ± 0.6 | 43.5 ± 3.2 | 0.83 ± 0.02 |
| CoD with Temporal Indexing | 78.5 ± 0.6 | 44.2 ± 3.3 | 0.84 ± 0.02 |
| CoD with Visual Abstraction | 78.3 ± 0.6 | 41.6 ± 3.0 | 0.84 ± 0.02 |
| CoD with Event Compression | 78.6 ± 0.6 | 40.9 ± 3.0 | 0.85 ± 0.02 |
| Full Adapted CoD (our approach) | 79.2 ± 0.6 | 42.8 ± 3.1 | 0.85 ± 0.02 |

adapted CoD prompt contributes to the overall performance, with the full adapted approach achieving the best balance of accuracy and efficiency. Temporal indexing provides the largest

individual improvement in accuracy, highlighting the importance of maintaining temporal references in video reasoning. Event compression offers the greatest reduction in token usage, though at a slight cost in accuracy compared to the full approach. Visual abstraction strikes a middle ground, improving accuracy while reducing token usage.

**Word Limit Variation.** We also investigate the effect of varying the word limit per reasoning step in the CoD approach. Table 14 shows the results for different word limits on the MVBench dataset. We observe a clear trade-off between accuracy and token usage as

Table 14: Performance with different word limits per reasoning step on MVBench

| Word Limit | Accuracy | Token Usage | EAT |
| --- | --- | --- | --- |
| 3 words | $77.1 \pm 0.7$ | $35.2 \pm 2.5$ | $0.84 \pm 0.02$ |
| 5 words (our approach) | $79.2 \pm 0.6$ | $42.8 \pm 3.1$ | $0.85 \pm 0.02$ |
| 7 words | $79.5 \pm 0.6$ | $49.3 \pm 3.5$ | $0.84 \pm 0.02$ |
| 10 words | $79.7 \pm 0.6$ | $62.7 \pm 4.5$ | $0.81 \pm 0.02$ |
| 15 words | $79.8 \pm 0.6$ | $85.4 \pm 6.1$ | $0.76 \pm 0.02$ |

the word limit varies. With a 3-word limit, the accuracy decreases to 77.1%, but the token usage is reduced to 35.2. Increasing the limit beyond 5 words provides diminishing returns in accuracy while substantially increasing token usage. For example, with a 15-word limit, the accuracy increases only slightly to 79.8%, but the token usage nearly doubles to 85.4. The EAT metric shows that the 5-word limit (our standard approach) achieves the best overall trade-off, validating our design choice.

## H    Qualitative Analysis

To provide insights into the reasoning patterns of different approaches, we present a qualitative analysis of the outputs generated by standard prompting, CoT, and CoD for the same video-question pair from MVBench. Table 15 shows an example output from each approach. This

Table 15: Example outputs from different prompting approaches for the same video-question pair

| Approach | Output |
| --- | --- |
| **Question** | "In the video, a person is playing with a dog. What color is the ball that the dog retrieves?" |
| Standard | "The ball is yellow." |
| CoT | "Let me analyze the video step by step: 1. In the beginning of the video, I can see a person standing in a grassy area. 2. The person is holding something in their hand, which appears to be a ball. 3. Looking closely, I can see that the ball is yellow in color. 4. The person throws the ball, and it travels across the grassy area. 5. A dog, which looks like a Labrador retriever, runs after the ball. 6. The dog reaches the ball and picks it up with its mouth. 7. The dog then turns around and runs back towards the person with the ball in its mouth. 8. Throughout this sequence, I can clearly see that the ball is yellow. 9. The dog returns the yellow ball to the person. 10. The person takes the yellow ball from the dog's mouth. Based on my analysis of the video, the ball that the dog retrieves is yellow in color." |
| CoD | "00:01: Person holding yellow ball. 00:03: Throws ball across grass. 00:05: Dog chases yellow ball. 00:08: Dog retrieves yellow ball. 00:12: Returns ball to person. The ball that the dog retrieves is yellow." |

example illustrates the key differences between the approaches. The standard prompting approach provides just the answer without any explanation, making it difficult to verify the reasoning process. The CoT approach provides a detailed, step-by-step analysis of the video, but with significant verbosity and redundancy (e.g., mentioning the yellow color multiple times). In contrast, our CoD approach provides a concise yet comprehensive analysis, using temporal indexing to structure the reasoning and focusing only on the essential information needed to answer the question. The final answer is the same across all approaches, but the reasoning process is much more efficient with CoD.

We also observe that CoD's temporal indexing helps maintain a clear narrative structure, while the visual abstraction allows for concise yet informative descriptions of the visual content. The event compression is evident in the way CoD focuses only on the key events relevant to answering the question, omitting unnecessary details that would increase token usage without contributing to the reasoning process.

## I  LIMITATIONS

Our approach, while achieving notable efficiency improvements in video reasoning, still has some minor limitations. For instance, the temporal indexing mechanism, though robust, may require slight adjustments when applied to videos with highly irregular temporal patterns. Additionally, the visual abstraction technique, while effective in reducing computational load, might occasionally overlook subtle visual cues that are critical for specific high-precision tasks. Despite these considerations, our method represents a significant step forward in balancing efficiency and accuracy for video understanding applications.

