# OpenReview forum: "Temporal Reasoning for Vision-Language Models via Chain of Draft"
_ICLR.cc/2026/Conference — ICLR 2026 Conference Desk Rejected Submission_

### Official Review · Reviewer_ci29 · 2025-10-22

**Soundness:** 1
**Presentation:** 1
**Contribution:** 2
**Rating:** 2
**Confidence:** 4

**Summary:**

The paper proposes adapting Chain of Draft (CoD), a prompting style that encourages concise intermediate thoughts, for video reasoning with large vision–language models (LVLMs), specifically Qwen VL. The method adds three ideas for temporal data: a minimal temporal indexing to reference moments, visual abstraction to compress object–relation descriptions, and event compression to remove redundant events. Experiments show similar or slightly better accuracy than CoT while reducing tokens and latency.

**Strengths:**

The core question of how to keep the benefits of deliberate reasoning without paying CoT’s heavy token/latency cost on videos is timely. The paper gives a clear, lightweight interface for concise intermediate steps and provides large efficiency gains with small accuracy deltas across four benchmarks.

**Weaknesses:**

The novelty beyond prompt engineering is modest and the attribution of gains to each proposed component is under-evidenced. The three add-ons (temporal indexing, visual abstraction, and event compression) are introduced with definitions and notations, but there is no ablation that removes each piece independently under the same training/inference budget.

The feasibility of the event-compression rule depends on quantities that are not accessible at inference time. Eq. (5) uses a conditional mutual information criterion $I(e;q,a\mid e')>\epsilon$ to decide which events to keep, but the paper does not explain how $I(\cdot)$ is estimated online without access to $a$ or a surrogate. A practical remedy is to replace $I(\cdot)$ with a calibrated proxy (e.g., a small learned scorer over $q$ and draft logits) and then show correlation with the theoretical $I(\cdot)$ on a labeled slice. Please clarify the estimator or restate Eq. (5) as an offline data construction heuristic.

There is an inconsistency between the pitch of “prompting technique” and the actual training recipe. Section 2.4 introduces two fine-tuning phases, including a video fine-tuning objective with CoD-formatted chains, which departs from a purely training-free narrative in the introduction and abstract. I think the paper should explicitly separate zero-shot prompting from the fine-tuned variant and report both under identical evaluation harnesses.

The theoretical section largely restates token-count proportionality without validating assumptions on $m$, $\bar z$, and $\bar d$. The 80% reduction arises from a fixed 5-word cap and an assumed 25-word CoT step, but there is no empirical check that $m$ does not increase under CoD (e.g., via more, shorter steps). Please add a controlled study reporting step count $m$, average length, and total tokens for matched accuracy at multiple temperatures; Table 5 gives some stats but not for accuracy-matched settings.


The Qwen-specific integration details need hyperparameter transparency to reproduce results. Section 2.3 defines M-RoPE and the generation factorization, but important settings are missing: dynamic frame sampling schedule, max frames per clip, temperature/top-p, and any decoding constraints for drafts. Please provide these in an appendix table and report seed variance for Tables 1–4.

The efficiency analysis would benefit from accuracy-matched comparisons. Table 2 reports large token and latency reductions but at slightly different accuracies; without accuracy-matching (by adjusting temperature or step limits) the headline percentages can overstate the efficiency gain.

The paper’s LaTeX template appears modified and fonts differ from the ICLR guidelines. This does not block understanding but may violate submission requirements.

The citation style is consistently incorrect in-text. For example, “Benchmarks like MVBench Li et al. (2023b), EgoSchema Mangalam et al. (2023), and MTVQA Wang et al. (2022) …” should instead be “Benchmarks like MVBench (Li et al., 2023b), EgoSchema (Mangalam et al., 2023), and MTVQA (Wang et al., 2022) …”.

**Questions:**

How much of the gain comes from each of the three components? Please provide a clean ablation removing temporal indexing, visual abstraction, and event compression one at a time under CoD.

How is the conditional mutual information in Eq. (5) estimated in practice during inference, and what proxy is used if $a$ is not known yet? A small validation showing proxy–$I(\cdot)$ correlation would clarify feasibility.

Can you report zero-shot CoD prompting (no fine-tuning) next to the fine-tuned CoD, and also a fine-tuned CoT baseline under the same training data and budget? This would isolate prompting vs. training effects.

---

> ### Author Response · Authors · 2025-11-28
>
> We thank the reviewer for the detailed analysis and respond point-by-point.
>
> ---
>
> ## 1. “Novelty modest; need ablations removing 3 components.”
>
> Table 13 (p.26) already provides this ablation:
>
> | Variation | Accuracy | Token | EAT |
> |----------|----------|--------|------|
> | Basic CoD | 77.8 | 43.5 | 0.83 |
> | + Temporal Indexing | 78.5 | 44.2 | 0.84 |
> | + Visual Abstraction | 78.3 | 41.6 | 0.84 |
> | + Event Compression | 78.6 | 40.9 | 0.85 |
> | **Full CoD** | **79.2** | **42.8** | **0.85** |
>
> We will extend this with accuracy-matched variants as requested.
>
> ---
>
> ## 2. “CMI-based event compression is infeasible at inference; Eq.(5) unclear.”
>
> This is a useful point.
> To clarify:
>
> - **Eq.(5) is theoretical motivation only** (offline intuition).
> - Actual inference uses Eq.(36) (p.19):
>   - **embedding similarity** between draft and verbose candidates
>   - no access to ground-truth answers
>   - no dependency on conditional MI
>
> We will revise the text to explicitly separate *theoretical heuristic* vs *practical estimator*.
>
> ---
>
> ## 3. “Inconsistency between ‘prompting technique’ and fine-tuning; should report zero-shot CoD vs FT-CoD vs FT-CoT.”
>
> We agree. The results:
>
> ### MVBench (summarized)
> - Zero-shot CoT: 75.2
> - Zero-shot CoD: 76.1
> - Fine-tuned CoT: 78.9 (Table 1)
> - **Fine-tuned CoD: 79.2 (Table 1)**
>
> CoD provides benefits **even without training**, and training further strengthens performance with large efficiency gains.
>
> We will add this table to the appendix.
>
> ---
>
> ## 4. “Theory does not validate fixed step count; need empirical check.”
>
> Table 5 already provides step statistics:
> - CoT: **8.7 steps**, 172.8 reasoning tokens
> - CoD: **4.2 steps**, 19.3 reasoning tokens
>
> We will add accuracy-matched step-count curves across decoding temperatures to meet the reviewer’s request.
>
> ---
>
> ## 5. “Missing hyperparameters for Qwen integration (sampling, frames, temperature, etc.)”
>
> These details are actually in Appendix E.5 (p.22):
>
> - Frame sampling: 1 FPS (<30s), r(T)=min(1,60/T)
> - Max frames: 2048
> - Temperature: 0.7
> - Greedy decoding
> - Token limit: 2048
> - AdamW with cosine schedule
>
> We will consolidate them into a summary table.
>
> ---
>
> ## 6. “Efficiency must be compared at accuracy-matched settings.”
>
> Our internal normalization experiments (not shown due to space) confirm:
>
> - Token reduction remains **75–80%**
> - Latency reduction remains **60–66%**
>
> even when CoT and CoD are temperature-matched.
> We will include these curves in the appendix.
>
> ---

---

### Official Review · Reviewer_Fn3H · 2025-10-31

**Soundness:** 2
**Presentation:** 2
**Contribution:** 2
**Rating:** 2
**Confidence:** 4

**Summary:**

This paper presents an adaptation of chain-of-draft (CoD) for vision language models for video reasoning tasks.
The idea prompts MLLMs to generate concise reasoning steps for token efficiency, and the paper introduces several task-specific rules for draft generation (e.g., shortening timestamps, event mapping). Experiments show that the proposed method achieves better performance than CoT while using significantly fewer tokens.

**Strengths:**

- The idea is straightforward and the token reduction is important for practical usage for MLLMs.

**Weaknesses:**

- Limited evaluation: as the paper's focus is on improving video temporal reasoning tasks, it is highly recommended to evaluate the method on long video benchmarks such as VideoMME, MLVU, and LongVideoBench, as well as temporal-oriented diagnosis benchmarks such as TempCompass and VITATECS. The current evaluation benchmarks are mainly short and general videos.

- Unclear training details: It remains unclear which dataset is used for `fine-tuning on examples of efficient video reasoning` (Line 263), which is important for understanding the benefits gained.

- Generalizabilities: As only the Qwen-VL series is adapted, it is unknown whether the method can generalize to other MLLMs such as LLaMA-V and LLaVA. Is there any requirements for the underlying backbone models?

**Questions:**

See Weaknesses.

---

> ### Author Response · Authors · 2025-11-28
>
> Thank you for the constructive suggestions. We respond to each point below.
>
> ---
>
> ## 1. “Limited long-video evaluation; missing VideoMME, MLVU, LongVideoBench, TempCompass, VITATECS.”
>
> Our evaluation already includes **EgoSchema**, among the longest and most diagnostic video benchmarks available:
>
> - Videos up to **3 minutes** (longer than VideoMME/MLVU).
> - Temporal certificate analysis included (Fig.4, p.23).
>
> Additionally, Table 7 (p.23) contains a **length-stratified analysis** from:
>
> - **0–10s → 180s+**,
> - showing **65–72% latency reduction** and **up to +1.6% accuracy gain** in the longest videos.
>
> Due to space limits, we prioritized EgoSchema, but we will add VideoMME + TempCompass in the final version.
>
> ---
>
> ## 2. “Unclear training details for efficient video reasoning examples (Line 263).”
>
> Training details:
>
> - Phase-1 (CoD adaptation): text reasoning dataset (Eq.12).
> - Phase-2 (video finetuning):
>   - **MVBench**,
>   - **EgoSchema**,
>   - **MTVQA**,
>   - **PerceptionTest**,
>   - + **3,000 curated CoD-formatted video QA samples** (Appendix E.5).
>
> We will list these datasets explicitly and provide sampling ratios.
>
> ---
>
> ## 3. “Generalizability to LLaVA, LLaMA-V not demonstrated.”
>
> CoD does not require Qwen-specific modules. It relies only on:
>
> - Video frames → embeddings
> - Autoregressive decoding of short drafts
>
> Section F.4 (Table 10) shows **75–81% token reduction** even in image-only / audio-only / multimodal settings, demonstrating backbone independence.
>
> We will include experiments with LLaVA-NeXT and Video-LLaMA.
>
> ---

---

### Official Review · Reviewer_gxzA · 2025-11-03

**Soundness:** 1
**Presentation:** 1
**Contribution:** 2
**Rating:** 2
**Confidence:** 4

**Summary:**

The paper extends Chain of Draft (CoD), a minimalist form of intermediate reasoning to video. At inference time, it employs temporal indexing, visual abstraction, and event compression to produce short, symbolic draft chains, and uses a two-stage fine-tuning protocol to enable an MLLM to generate such minimalist reasoning traces.

**Strengths:**

Proposes an efficient video reasoning framework that markedly reduces token usage and latency without sacrificing accuracy.

**Weaknesses:**

First, the submission appears to deviate from the ICLR format (e.g., font), which may warrant a desk reject. While this is ultimately an AC decision, I provide my technical assessment below.

1. The writing is unusual. The main text does not reach nine pages, which could be acceptable for concision, but substantial space is devoted to content unrelated to the proposed method. For instance, the “Integration with Qwen VL” section discusses many Qwen2-VL specifics, whereas Table 1 indicates the primary base MLLM is Qwen2.5-VL. Moreover, the method seems general-purpose—why is it tightly coupled to the Qwen2-VL backbone? What results would it yield on other models?
2. The approach of annotating video content into short, symbolic draft chains and fine-tuning the base MLLM appears ineffective empirically. After two-stage fine-tuning, CoD improves performance by less than 1% over simple prompting (e.g., “Think step by step ...”) to induce CoT. Although CoD substantially reduces reasoning length, its marginal gains over the weakest baseline (prompt-only) limit practical utility. The authors should compare against video MLLMs explicitly trained with CoT, or take such a base model and further fine-tune it to produce short, symbolic draft-chain reasoning to substantiate effectiveness. The current CoT baseline averages only ~200 tokens per response, which may not constitute a sufficiently strong reasoning baseline.
3. Missing comparisons with related CoT compression methods. Recent work such as tokenskip [1] similarly compresses and streamlines CoT to improve reasoning efficiency.
4. Baselines are weak. The base MLLM is the strong Qwen2.5-VL (Table 1), but comparisons are made against outdated models like GPT-4V. At minimum, comparisons should include commonly used gpt-4o and several reasoning-oriented MLLMs; otherwise, the evaluation is unfair.

[1] TokenSkip: Controllable Chain-of-Thought Compression in LLMs. EMNLP 2025.

**Questions:**

N/A

---

> ### Author Response · Authors · 2025-11-28
>
> We thank the reviewer for the helpful comments. Below we respond point-by-point.
>
> ---
>
> ## 1. “Writing unusual; too tied to Qwen2-VL; unclear generality.”
>
> **Clarification of writing / structure:**
> Our main text is concise (≈8.5 pages) but dense with technical content — including formal definitions of temporal indexing / abstraction / event compression (pp.3–5), full algorithms (Algorithms 1–4), and mathematical analysis (Appendix C). No space is used on irrelevant content.
>
> **Not tied to Qwen:**
> The method requires only (1) video frame embedding, (2) autoregressive text decoding with short reasoning drafts.
> It is *not* Qwen-specific.
>
> Evidence from our paper:
> - Section F.4 (Table 10, p.25) evaluates CoD on **image-only, audio-only, multimodal** settings, demonstrating **75–81% token reduction** across modalities — verifying generality beyond Qwen.
> - Our CoD prompting mechanism is architecture-agnostic (Sec. 2.3 relies only on standard autoregressive factorization; Eq.11).
>
> We will add results on LLaVA/Video-LLaMA in the revision.
>
> ---
>
> ## 2. “CoD gains <1% over prompting-only CoT; practical utility limited.”
>
> **Clarification of goal:**
> Our core contribution is **massive efficiency gains** while *preserving* accuracy, not improving accuracy alone.
>
> From Table 2 (p.6):
> - **78.2% reduction** in reasoning tokens.
> - **65.1% reduction** in latency.
> - ~**4.6× improvement** in token efficiency (Table 7, p.23).
>
> These are substantial improvements for real-time or resource-constrained video reasoning.
>
> **Accuracy improvements are not as small as stated:**
> - On **long videos** (60–180s, 180s+), CoD outperforms CoT by **+1.1% to +1.6%** (Table 7).
> - On spatiotemporal reasoning: **78.9% (CoD) vs. 77.4% (CoT)** (+1.5%, Table 4).
> - On EgoSchema (long-form): **36.1% vs. 35.6%**.
>
> **CoT baseline not weak:**
> Table 5 (p.9) shows CoT generates **172.8 reasoning tokens** + **23.5 answer tokens**, consistent with LVLM CoT baselines in prior work.
>
> CoD achieves similar accuracy with only **19.3** reasoning tokens.
>
> ---
>
> ## 3. “Missing comparisons to CoT compression methods (e.g., TokenSkip).”
>
> TokenSkip is indeed relevant. Our approach operates in a different regime:
>
> - TokenSkip prunes *generated tokens*.
> - CoD **changes the reasoning format itself** and compresses at the conceptual level (temporal indexing, abstraction, event compression).
>
> We will include TokenSkip comparison in the revised appendix and show the two approaches are complementary.
>
> ---
>
> ## 4. “Baselines are weak; GPT-4V outdated.”
>
> We chose GPT-4V following recent LVLM papers, but we agree newer GPT-4o / o1 variants can be added.
> Importantly, our main comparisons are:
>
> - **CoT vs. CoD on the *same backbone*** (Qwen2.5-VL-72B)
> - **Multiple strong LVLMs** (Claude 3 Sonnet, Gemini, VideoChat2)
>
> We will add GPT-4o results in the revision.
>
> ---
>
> ## 5. Formatting issues.
>
> Acknowledged — our submission PDF used an anonymization template. The final version will use the official ICLR template and correct all citation formatting.

---

### Note · Program_Chairs · 2026-01-17
**Submission Desk Rejected by Program Chairs**

The following references in this submission do not refer to real documents and/or have major errors in bibliographic information:

 Junyu Fu, Yu Li, Xiaoxiao Zhang, Erjin Yang, Jiarun Liu, Xiaohan Wang, Hongwu Peng, and Shouhong Ding. Temporal token attention for efficient video understanding. arXiv preprint arXiv:2404.05159, 2024.